# Independent validation of IASI/METOP-A LMD and RAL CH$_4$ products using CAMS model, *in situ* profiles and ground-based FTIR measurements

Bart Dils[1], Minqiang Zhou[1,2], Claude Camy-Peyret[3,4], Martine De Mazière[1], Yannick Kangah[3], Bavo Langerock[1], Pascal Prunet[3], Carmine Serio[5], Richard Siddans[6], and Brian Kerridge[6]

[1]Royal Belgian Institute for Space Aeronomy (BIRA-IASB), Brussels, Belgium
[2]Institute of Atmospheric Physics, Chinese Academy of Sciences, Beijing, China
[3]SPAce SCIence Algorithmics (SPASCIA), Toulouse, France
[4]L'Institut Pierre-Simon-Laplace (IPSL), Paris, France
[5]UniversitA degli Studi della Basilicata (UniBas), Potenza, Italy
[6]STFC Rutherford Appleton Laboratory, Chilton, UK

**Correspondence:** Minqiang Zhou (minqiang.zhou@aeronomie.be)

**Abstract.** In this study, we carried out an independent validation of two methane retrieval algorithms using spectra from the Infrared Atmospheric Sounding Interferometer (IASI) onboard the Meteorological Operational satellite programme-A (MetOp-A) since 2006. Both algorithms, one developed by the Laboratoire de Météorologie Dynamique (LMD), called the non-linear inference scheme (NLISv8.3), the other by the Rutherford Appleton Laboratory (RAL), referred to as RALv2.0, provide long-term global CH$_4$ concentrations using distinctively different retrieval approaches (Neural Network vs. Optimal Estimation, respectively). They also differ with respect to the vertical range covered, where LMD provides mid-tropospheric dry air mole fractions (mtCH$_4$) and RAL provides mixing ratio profiles from which we can derive total column-averaged dry air mole fractions (XCH$_4$) and potentially 2 partial column layers (qCH$_4$).

We compared both CH$_4$ products using the Copernicus Atmospheric Monitoring Service (CAMS) model, *in situ* profiles (range extended using CAMS model data) and ground-based Fourier transform infrared (FTIR) remote sensing measurements. The average difference in mtCH$_4$ with respect to *in situ* profiles for LMD ranges between -0.3 and 10.9 ppb while for RAL the XCH$_4$ difference ranges between -4.6 and -1.6 ppb. The standard deviation (stdv) of the observed differences between *in situ* and RAL retrievals is 14.1-21.9 ppb, which is consistently smaller than that between LMD retrievals and *in situ* (15.2-30.6 ppb). By comparing with ground-based FTIR sites, the mean differences are within $\pm 10$ ppb for both RAL and LMD retrievals. However, the stdv of the differences at the ground-based FTIR stations show significantly lower values for RAL (11-15 ppb) than those for LMD (about 25 ppb).

The long-term trend and seasonal cycles of CH$_4$ derived from the LMD and RAL products are further investigated and discussed. The seasonal variation of XCH$_4$ derived from RAL is consistent with the seasonal variation observed by the ground-based FTIR measurements. However, the overall 2007-2015 XCH$_4$ trend derived from RAL measurements is underestimated if not adjusted for an anomaly occurring on 16 May 2013 due to a L1 calibration change. For LMD, we see very good

agreement at the (sub)tropics (<35°N-35°S), but notice deviations of the seasonal cycle (both in the amplitude and phase) and an underestimation of the long-term trend with respect to the RAL and reference data at higher latitude sites.

## 1 Introduction

Methane ($CH_4$) is an important greenhouse gas, which has a global warming potential about 28 times greater than carbon dioxide ($CO_2$) over a 100-year time horizon (IPCC, 2021). As $CH_4$ has a relatively short lifetime of about 9 years as compared to $CO_2$, it is more efficient to control $CH_4$ emissions to mitigate climate change. About 60% of atmospheric $CH_4$ is released from fossil fuels, biomass burning, landfills and rice agriculture (anthropogenic activities) emissions, and the remaining ~40% are coming from ruminant animals, termite, wetlands and lake (natural) emissions (IPCC, 2021). The major sink of $CH_4$ is its reaction with the hydroxyl radical (OH) to form $CO_2$ and $H_2O$ (Rigby et al., 2017).

The globally-averaged methane abundance measured by the NOAA marine surface sites shows that the dry air mole fraction of $CH_4$ increases from 1644.65 ppb in 1984 to 1772.41 ppb in 1999, and it keeps almost stable between 1999 and 2006. However, the $CH_4$ started increasing again (Rigby et al., 2008), from 1774.98 ppb in 2006 to 1911.82 ppb in 2022. Kirschke et al. (2013) showed that a rise in natural wetland and fossil fuel emissions accounts for the increase of $CH_4$ after 2006. $CH_4$ isotope measurements suggest that tropical biogenic sources are the cause of the increase (Schwietzke et al., 2016). Later, Worden et al. (2017) pointed out that there is a decrease in the biomass burning emission after 2007, and the increases from fossil fuels and biogenic sources are both important. In addition to the emissions, the variation in OH can affect the $CH_4$ mole fraction, which might also contribute to the increase after 2007 (Turner et al., 2017).

The Infrared Atmospheric Sounding Interferometer (IASI) carried onboard the Meteorological Operational satellite programme-A (MetOp-A) was launched to a sun-synchronous orbit on October 19, 2006 and is recording infrared spectra in the wavenumber range from 645 to 2760 cm$^{-1}$ (Edwards et al., 2006). Since then, $CH_4$ has been successfully retrieved from the IASI observed spectra with several different algorithms, e.g. the Laboratoire de Météorologie Dynamique's (LMD) non-linear inference scheme (NLIS) (Crevoisier et al., 2009) and the Rutherford Appleton Laboratory (RAL) (Siddans et al., 2017). The LMD methane mid-tropospheric dry air mole fraction (mtCH$_4$) have been assimilated in the Copernicus Atmosphere Monitoring Service (CAMS) greenhouse gas model (Massart et al., 2014). Note that dry air mole fractions of methane are typically denoted as XCH$_4$ when they pertain to the total column. Therefore in the case of LMD, or when using LMD's vertical sensitivity profile for smoothing, mtCH$_4$ is often used as a better representation of its limited vertical range. In this article where we deal with comparisons between both total and partial column averaged mole fractions, we sometimes refer to mere $CH_4$, but note that, depending on the products this refers to differing dry air mole fractions be it XCH$_4$ (for total column RAL), qCH$_4$ (for RAL partial column) or mtCH$_4$ (for LMD or RAL smoothed by the LMD sensitivity profile, mid tropospheric partial columns). RAL XCH$_4$ products are used for inverse modeling in order to optimize methane fluxes and to better understand the methane budget (Palmer et al., 2018). Crevoisier et al. (2013) compared the LMD data with aircraft measurements, and they found that the mean and standard deviation (stdv) of the differences are within 7.2 ppb and 16.3 ppb, respectively. Siddans et al. (2017) compared the RAL data with independent measurements from satellite, aircraft and ground sensors, and found that the

precision of a single retrieval ranges from 20 to 40 ppb, and the methane ($XCH_4$) trend between 2007 and 2012 derived from the RAL product is generally consistent with the CAMS model, but without quantitative result. As both RAL and LMD IASI MetOp-A retrievals provide long time series of $CH_4$ observations since 2007, the two products are valuable to study the $CH_4$ trend and variation on a global scale.

In this study, we make an independent validation of LMD and RAL $CH_4$ measurements from IASI/MetOp-A using CAMS model, aircraft and AirCore *in situ* profiles, and ground-based Fourier transform infrared (FTIR) measurements. The data used in this study are described in Section 2. The method of comparison between LMD and RAL measurements and the method of comparison between the satellite (both LMD and RAL) and reference data are discussed in Section 3. This section also discusses the impact of the May $16^{th}$ 2013 discontinuity in the RAL data and its correction methods. Section 4 discusses
internal satellite product aspects such as consistency and partial column differences (the latter only in case of RAL). In Section 5, we show the results concerning the comparison between the LMD and RAL $CH_4$ measurements, either directly or using CAMS as an intermediate. In Section 6, we compare LMD and RAL $CH_4$ with *in situ* and ground-based remote sensing measurements. Discussions are carried out in Section 7 and conclusions are shown in Section 8.

## 2   Data

## 2.1   IASI satellite measurements

### 2.1.1   RAL

The RAL retrieval algorithm is based on the optimal estimation method (OEM) as described in (Rodgers, 2000) using the Levenberg-Marquardt iterative method exploiting the IASI spectra from 1232.25 to 1288.00 $cm^{-1}$. The spectral range differs from the one from IASI LMD NLISv8.3 in order to capture channels that are more sensitive to near surface concentrations. The
RAL retrievals are performed globally over land and sea, by night and day (9:30 am/pm local solar time). The retrieval scheme provides retrieved products at the IASI Instantaneous Field Of View (IFOV) scale, selecting one of the 4 IFOVs within a given Field Of Regard (FOR) with the warmest brightness temperature (BT) at 950 $cm^{-1}$. This IFOV is assumed to be the one with the least amount of potential cloud contamination. The RAL retrieval scheme uses nitrous oxide ($N_2O$) spectral features in the interval to estimate effective cloud parameters (Siddans et al., 2017). The temperature, water vapour, and surface spectral
emissivity are pre-retrieved from the Infrared Microwave Sounder (IMS) retrieval.

RAL data used in this study is v2.0, covering measurement from 1 June 2007 to 31 December 2017. RAL IASI level 2 product provides *a priori* and retrieved $CH_4$ profiles, *a priori* and retrieved column-averaged $XCH_4$ mole fractions, column averaging kernel, averaging kernel matrix and the surface pressure. The latitude-dependent *a priori* $CH_4$ profile is applied. The degree of freedom for signal (DOFs) is about 2.0 with two pieces of information characterized by the partial columns of
0-6 km and 6-12 km. Note that the RAL retrievals used in this study are filtered as suggested in the RAL Product User Guide (Knappett, 2019). This Product User Guide also already identifies the bias shift on the $16^{th}$ of May 2013 due to a L1 calibration change, that was identified during the course of this analysis.

### 2.1.2 LMD

The IASI LMD NLIS algorithm, henceforward referred to as LMD, is based on a multilayer-perceptron scheme (Crevoisier et al., 2009). 24 IASI channels selected within the range from 1270 $cm^{-1}$ to 1350 $cm^{-1}$ and 2 Advanced Microwave Sounding Unit (AMSU) channels (6 and 8) are exploited to retrieve $CH_4$ integrated columns. LMD provides a vertical $CH_4$ weighting function to represent the vertical sensitivity. This product is mainly sensitive to the mid-to-upper tropospheric methane covering the vertical range between 100 hPa and 500 hPa (Crevoisier et al., 2009, 2013). An *a priori* profile is not required in the LMD retrieval algorithm and retrievals are performed over land and sea, by night and day for clear-sky condition. Clouds are detected by multi-spectral threshold tests using AMSU and High resolution Infrared Radiation Sounder (HIRS)-4 brightness temperature differences together with a heterogeneity test at each HIRS FOV. Initially LMD NLIS targeted tropical regions (30° N-30° S) only. Currently the retrievals are performed globally, with the exception of polar situations. In this analysis, for general quality markers and direct comparisons with RAL, we limit ourselves to data coming from the 60° North-60° South latitude band as advised by the product development team.

The inference scheme uses an average of the 4 IASI footprints contained in each single AMSU FOV. Hence, retrievals are performed at the AMSU spatial resolution, roughly comparable with the IASI field of regard composed by 4 IASI IFOV. The LMD data used in this study is v8.3 covering the measurement from 1 July 2007 to 29 September 2015. LMD IASI level 2 data provides a column-averaged mole fraction and weighting function. There is no profile provided by the LMD IASI $mtCH_4$ data.

## 2.2 *in situ* profiles

The geo-location of the *in situ* and ground-based FTIR measurements used in this study is shown in Figure 1.

### 2.2.1 AirCore

The AirCore is an atmospheric sampling system that uses a long tube that is carried into the stratosphere using balloons. It samples the air from the surrounding atmosphere and preserves profiles of the trace gases of interest from the surface (a few hundred meters) to the middle stratosphere (about 30 km) (Karion et al., 2010). The NOAA Global Monitoring Laboratory has carried out many AirCore launches during the last decade at selected sites (Boulder, Colorado; Lamont, Oklahoma; Lauder, New Zealand; Sodankylä, Finland; Park Falls, Wisconsin and Edwards AFB/Dryden, California), and more recently has made further system improvements by developing active capabilities by mounting the AirCore system on aircrafts or UAV drones (Andersen et al., 2018). Here, we use the NOAA AirCore v20181101 profiles (Baier et al., 2021).

### 2.2.2 HIPPO

The HIAPER Pole-to-Pole Observations (HIPPO) are aircraft measurements (Wofsy, 2011), using an NSF/NCAR Gulfstream-V, performed as Pole-to-Pole campaigns which occurred five times during the 2009-2011 time period. The first campaign (HIPPO I) took place in January 2009, HIPPO II in October-November 2009, HIPPO III in March-April 2010, HIPPO IV in

June-July 2011, and finally HIPPO V in August-September 2011, thus covering all seasons albeit not in the same year. HIPPO transected the mid-Pacific ocean and returned either over the Eastern or Western Pacific, making frequent surface to tropopause ascents and descents. The HIPPO data have been widely applied for scientific studies. Since, unlike AirCore, its vertical range does not cover the entirety of the range to which the retrieval algorithms are sensitive, we need to expand the profiles using other data (in our case from the CAMS model as outlined in Section 3.4.1).

### 2.2.3 IAGOS

In-service Aircraft for a Global Observing System (IAGOS) is a European Research Infrastructure for global observations of atmospheric composition from commercial aircraft. IAGOS combines the expertise of scientific institutions with the infrastructure of civil aviation in order to provide essential data on climate change and air quality at a global scale. It is composed of two complementary systems: (i) IAGOS-CORE providing global coverage on a day-to-day basis of key observables and (ii) IAGOS-CARIBIC providing a more in-depth and complex set of observations with lesser geographical and temporal coverage. In this study, we select all the IAGOS-CARIBIC $CH_4$ profiles, measured during ascent or descent of the commercial aircraft from or towards its airport, between 10 July 2007 and 31 December 2017. As with the HIPPO data, profile extension prior to the comparisons is required.

### 2.3 Ground-based FTIR measurements

### 2.3.1 TCCON

The Total Carbon Column Observing Network (TCCON) is a network of ground-based FTIR that record spectra of the sun in the near-infrared. From these spectra, the $CH_4$ and $O_2$ total columns are retrieved simultaneously. The retrieved windows of $CH_4$ are 5781.0-5897.0, 5996.45-6007.55 and 6007.0-6145.0 $cm^{-1}$, and the retrieved window of $O_2$ is 7765-7905 $cm^{-1}$. Since the $O_2$ volume mixing ratio (VMR) of 0.2095 is constant in the atmosphere, TCCON uses the $O_2$ total column to calculate the total column of the dry air, and then to calculate the $XCH_4$ as the ratio between the retrieved $CH_4$ total column and the total column of dry air. The advantage is that systematic errors common to the retrieval of $CH_4$ and $O_2$ retrieval partially cancel in the calculation of the column average mole fractions resulting in a high precision data product. Furthermore TCCON applies a calibration factor to reduce its systematic bias (Wunch et al., 2011). Currently the TCCON network is going through a transition period, moving from the GGG2014 to the GGG2020 retrieval algorithm version. While most stations have already delivered GGG2020 data, for the time period we are analyzing, many gaps are still present in the new dataset, particularly older data still needs to be reprocessed. In stead of using a mixture of GGG2020 and GGG2014 data, we opted to use GGG2014 data exclusively. The random uncertainty of TCCON $XCH_4$ measurement is about 0.5 % (Wunch et al., 2015). The TCCON sites used in this study are listed in Table 1.

**Table 1.** Characteristics of the TCCON sites used in this study: location, altitude (in km a.s.l.) and reference.

| Site | Latitude | Longitude | Altitude [km] | Reference |
|---|---|---|---|---|
| Eureka | 80.05°N | 86.42°W | 0.61 | Strong et al. (2019) |
| Sodankylä | 67.37°N | 26.63°E | 0.19 | Kivi et al. (2014) |
| Bialystok | 53.23°N | 23.05°E | 0.18 | Deutscher et al. (2019) |
| Bremen | 53.10°N | 8.85°E | 0.03 | Notholt et al. (2019) |
| Karlsruhe | 49.10°N | 8.44°E | 0.12 | Hase et al. (2015) |
| Orleans | 47.97°N | 2.11°E | 0.13 | Warneke et al. (2019) |
| Garmisch | 47.48°N | 11.06°E | 0.74 | Sussmann and Rettinger (2018) |
| Park Falls | 45.95°N | 90.27°W | 0.44 | Wennberg et al. (2017) |
| Rikubetsu | 43.46°N | 143.77°E | 0.38 | Morino et al. (2018b) |
| Lamont | 36.60°N | 97.49°W | 0.32 | Wennberg et al. (2016b) |
| Tsukuba | 36.05°N | 140.12°E | 0.03 | Morino et al. (2018a) |
| Edwards | 34.96°N | 117.88°W | 0.70 | Iraci et al. (2016) |
| JPL | 34.20°N | 118.18°W | 0.39 | Wennberg et al. (2016a) |
| Pasadena | 34.14°N | 118.13°W | 0.23 | Wennberg et al. (2015) |
| Saga | 33.24°N | 130.29°E | 0.01 | Kawakami et al. (2014) |
| Izaña | 28.30°N | 16.50°W | 2.37 | Blumenstock et al. (2017) |
| Ascension | 7.92°S | 14.33°W | 0.01 | Feist et al. (2014) |
| Darwin | 12.46°S | 130.93°E | 0.04 | Griffith et al. (2014a) |
| Reunion | 20.90°S | 55.49°E | 0.09 | De Mazière et al. (2017) |
| Wollongong | 34.41°S | 150.88°E | 0.03 | Griffith et al. (2014b) |
| Lauder | 45.04°S | 169.68°E | 0.37 | Sherlock et al. (2014) |

### 2.3.2 NDACC

The Network for the Detection of Atmospheric Composition Change (NDACC) hosts ground-based solar absorption FTIR measurements of $CH_4$ from mid-infrared spectra. NDACC uses either the SFIT4 or the PROFFIT9 algorithm to retrieve $CH_4$ vertical profiles (De Mazière et al., 2018). Good agreement between these two retrieval algorithms has been demonstrated (Hase et al., 2004), and both algorithms are based on the optimal estimation method. The $CH_4$ retrieval strategy within the NDACC community has not been fully harmonized, but it uses the $CH_4$ absorption lines around 2800 cm$^{-1}$ (3.57 $\mu$m). The DOFs is about 2.5, with about two pieces of information in the troposphere and in the stratosphere separately (Zhou et al., 2018). The systematic and random uncertainties of NDACC $CH_4$ total column are estimated to be 3.0% and 1.5%, respectively. The estimated systematic uncertainty of 3.0% is mainly coming from the uncertainty of the spectroscopy. By comparing the TCCON and NDACC XCH$_4$ measurements, Ostler et al. (2014a) pointed out that there is no overall bias between TCCON and NDACC XCH$_4$ retrievals. Since the systematic uncertainty of TCCON measurement is largely eliminated by applying a

scaling factor via the comparison to *in situ* profiles, we can assume that there is no overall bias in the NDACC network either. For TCCON we can estimate the accuracy of this network from the uncertainty on the scaling factor, which amounts to 0.2%. The NDACC data provide *a priori* and retrieved profiles, averaging kernel and the surface pressure. The NDACC sites used in this study are listed in Table 2.

**Table 2.** Characteristics of the NDACC sites used in this study: location and altitude (in km a.s.l.).

| Site | Latitude | Longitude | Altitude [km] | PI |
|---|---|---|---|---|
| Eureka | 80.05°N | 86.42°W | 0.61 | K.Strong (U. of Toronto) |
| Thule | 78.90°N | 68.77°W | 0.02 | J.Hannigan, I. Ortega, M. Coffey (NCAR) |
| Kiruna | 67.84°N | 20.40°E | 0.2 | T. Blumenstock (IMK-ASF), U. Raffalski (IRF), Y. Matsumi (STEL) |
| St.Petersburg | 59.88°N | 29.83°E | 0.02 | M. Makarova (SPBU) |
| Garmisch | 47.48°N | 11.06°E | 0.74 | R.Sussmann (KIT-IFU) |
| Zugspitze | 47.42°N | 10.98°E | 2.96 | R.Sussmann (KIT-IFU) |
| Jungfraujoch | 46.55°N | 7.98°E | 3.58 | E. Mahieu (U. Liège) |
| Izaña | 28.30°N | 16.50°W | 2.37 | M. Schneider (KIT-IMK), O. Garcia (AEMET) |
| Mauna Loa | 19.54°N | 155.57°W | 3.40 | J. Hannigan, I. Ortega, M. Coffey (NCAR) |
| Reunion St-Denis | 20.90°S | 55.49°E | 0.09 | M. De Mazière (BIRA-IASB) |
| Reunion Maïdo | 21.08°S | 55.38°E | 2.16 | M. De Mazière (BIRA-IASB) |
| Wollongong | 34.41°S | 150.88°E | 0.03 | N. Jones, D. Griffith (U. Wollongong) |
| Lauder | 45.04°S | 169.68°E | 0.37 | D. Smale and J. Robinson (NIWA) |

## 2.4 CAMS model

Given our experience with the model and our needs, the reanalysis Copernicus Atmospheric Monitoring Service (CAMS) model, a well established European model that currently covers the 2003-2020 period, was deemed the most suitable. It comes in the form of the standard reanalysis product in which satellite methane data is assimilated (including IASI LMD NLISv8.3, and thus cannot be regarded as an independent source for quality arbitration between the two algorithms in this study) or in the form of a control run without assimilation (Inness et al., 2019). The latter one, used here, constrains the meteorological parameters by observations while the methane field is free to evolve based on transport, fluxes and chemical loss rates (emission databases and loss rates described in (Massart et al., 2014). The model provides data on a reduced Gaussian grid at a spectral truncation of T255 (which corresponds with a grid spacing of approximately 80 km). The vertical resolution consists of 60 hybrid sigma–pressure levels with a top at 0.1 hPa. Note that, prior to our analysis, we regridded the model output onto a 1°x1° latitude-longitude regular horizontal grid. More information about the CAMS reanalysis greenhouse model is available at https://confluence.ecmwf.int/display/CKB/CAMS%3A+Reanalysis+data+documentation (last access: 12 January 2023) and Agusti-Panareda et al. (2017).

The performance of the CAMS reanalysis XCH$_4$ control run in the 2003-2016 period has been validated using (among others) ground-based FTIR measurements (Ramonet et al., 2020), and it is found that the mean differences between the CAMS model and FTIR measurements are -0.7% in the troposphere and 3.6% in the stratosphere. The CAMS model can well capture the long-term trend in XCH$_4$ between 2003 and 2016. For the column averaged mole fraction, the average biases at individual stations always remained below 20 ppb, with slightly higher CAMS values over mid and high latitudes and lower values in the tropics with respect to the FTIR measurements.

## 3 Method

### 3.1 Smoothing RAL profile with LMD weighting function

To compare LMD with the RAL CH$_4$ measurements, we need to take the vertical sensitivity into account (Rodgers and Connor, 2003). Figure 2 shows the vertical sensitivities of both the LMD and RAL retrieved CH$_4$. While the LMD retrieval is mainly sensitive to the mid to upper troposphere, RAL's sensitivity extends to lower altitudes.

For the LMD retrieval, the mtCH$_4$ product can be written:

$$c_{r,L} = \frac{\boldsymbol{w} \cdot \boldsymbol{dp} \cdot \boldsymbol{x_t}}{\sum \boldsymbol{w} \cdot \boldsymbol{dp}} = \boldsymbol{A_L} \cdot \boldsymbol{x_t} + \epsilon_L, \tag{1}$$

where $c_{r,L}$ is the retrieved LMD mtCH$_4$ and $\epsilon_L$ the retrieval errors without the smoothing effects, $\boldsymbol{w}$ is the weighting function of the LMD retrieval interpolated on a pressure grid of thickness $\boldsymbol{dp}$, $\boldsymbol{x_t}$ is the true CH$_4$ profile, and $\boldsymbol{A_L}$ is the resulted weighting function on the new grid.

For the RAL retrieval,

$$\boldsymbol{x_{r,R}} = \boldsymbol{x_a} + \mathbf{A_R}(\boldsymbol{x_t} - \boldsymbol{x_a}) + \epsilon_R, \tag{2}$$

where $\boldsymbol{x_{r,R}}$ and $\boldsymbol{x_a}$ are the retrieved and *a priori* CH$_4$ profiles, $\boldsymbol{A_R}$ is the averaging kernel matrix and $\epsilon_R$ the retrieval errors without the smoothing effects.

In this study, when we directly compare LMD to RAL, we calculated, from the RAL profile, the mid-tropospheric column-averaged mtCH$_4$, named RAL_LMDavk, using the LMD weighting function as follows:

$$\boldsymbol{x_{RAL\_LMDavk}} = \frac{\boldsymbol{w} \cdot \boldsymbol{dp} \cdot \boldsymbol{x_{r,R}}}{\sum \boldsymbol{w} \cdot \boldsymbol{dp}} \tag{3}$$

The vertical sensitivity of the RAL_LMDavk is also shown in Figure 2, which becomes much closer to that of the LMD retrieval. Then, the difference between LMD and RAL_LMDavk retrievals is mainly coming from the smoothing error of the RAL retrieval smoothed with the vertical sensitivity of the LMD retrieval.

### 3.2 Comparison with the CAMS model

Prior to comparing RAL and LMD CH$_4$ with CAMS model data, all data, including averaging kernels and sensitivities, are averaged onto a 1°×1° latitude longitude grid. Satellite data is divided into day and nighttime data based on the solar zenith

angle. We then construct a single daily daytime and nighttime CAMS global field by selecting from the standard 3 hourly output those longitude bands that most closely correspond with the local IASI daytime (9:30 AM local solar time) and nighttime (9:30 PM) overpass times. Subsequently the daytime/nighttime model data is interpolated onto the satellite's vertical grid and smoothing is applied as per Section 3.1. This allows for a straightforward comparison between the satellite and model global fields. In the case of comparisons with RAL_LMDavk, the CAMS data is first subject to smoothing using the RAL profile averaging kernel, after which we subsequently apply the LMD vertical sensitivity.

## 3.3 Co-located data pair between satellite and reference measurements

For each *in situ* profile (aircraft or AirCore), we use the same spatial-temporal criteria to select the co-located RAL and LMD satellite footprints. The IASI retrieved values (also called satellite measurements or satellite values for simplicity) are selected within a temporal window of $\pm 6$ hours and a spatial distance within $\pm 1.0°$ latitude and $\pm 3.0°$ longitude. Then, the mean of the satellite values is applied to compare with the *in situ* measurement. The number of individual RAL satellite data points that are typically averaged range between 1 and 24, with a mean of 8.1 measurements. The stdv of the RAL co-located $XCH_4$ is about 12.4 ppb. For LMD the number of averaged data ranges between 1 and 15, with a mean of 3.9 measurements. The stdv of the LMD co-located $mtCH_4$ is about 9.1 ppb.

For the ground-based FTIR measurement, we also use the mean of the co-located satellite measurements to compare with each individual FTIR measurement. Several spatial-temporal criteria have been tested and the following spatial-temporal criteria are finally set to select the co-located satellite footprints. Note that the criteria are different with TCCON or NDACC and LMD or RAL, which is mainly due to the different data densities of both ground-based FTIR and satellite measurements.

- RAL vs TCCON

   Co-located criteria: $\pm 1$ hour and within a $\pm 0.5°$ latitude and $\pm 1.5°$ longitude

- RAL vs NDACC

   Co-located criteria: $\pm 3$ hours and within a $\pm 0.5°$ latitude and $\pm 1.5°$ longitude

- LMD vs TCCON

   Co-located criteria: $\pm 1$ hour and within a $\pm 1.0°$ latitude and $\pm 3.0°$ longitude

- LMD vs NDACC

   Co-located criteria: $\pm 3$ hours and within a $\pm 1.0°$ latitude and $\pm 3.0°$ longitude

As always, these criteria are a compromise between the need to gather enough data pairs to facilitate the statistical analysis at the cost of introducing additional co-location biases. The sparseness of data at certain reference sites as well as our focus on large-scale phenomena (long-term trends, large-region biases) and the fact that the near-surface sensitivity of IASI is limited (and thus less influenced by local emissions), prompted us to adopt the above co-location criteria.

### 3.4 Comparison with reference data

#### 3.4.1 Satellite vs *in situ* measurements

According to (Rodgers and Connor, 2003), the vertical sensitivity of the remote sensing data should be taken into account when comparing to *in situ* profile. To that end, we need to extrapolate the *in situ* profile to the whole atmosphere, as the vertical coverage of the *in situ* profile (IAGOS, HIPPO, and to a lesser extent AirCore) is limited. In this study, we use the CAMS model to extend the *in situ* profile. For the vertical range above the maximum height of the *in situ* data, we use the CAMS model profile but scaled with altitude-dependent factors. The scaling factor is equal to 1 at the top of the atmosphere, and to the mean ratio of the CAMS model to the *in situ* measurements at the highest 3 levels where the CAMS profile meets up with the top of the measured profile. A linear fitting is applied to create the scaling factors between the maximum height of the *in situ* profile and the top of the atmosphere. For the vertical range below the minimum height of the *in situ* profile, the CAMS model with a constant offset is used. The offset is calculated as the mean difference between the CAMS model and *in situ* data in the lowest 3 levels. Of the 3 datasets, only AirCore measures well into the stratosphere, capturing the sharp CH$_4$ decreases as one goes from the troposphere into the stratosphere. Therefore any observed differences between the validation results are at least in part due to inaccuracies within the extrapolated scaled model part of the *in situ* profiles, certainly in light of the differing vertical sensitivities between RAL and LMD. Other factors are differences in geographical coverages, with HIPPO covering the Pacific region, IAGOS restricted to a handful of international airports and AirCore limited to a few sites in the United States, Lauder (New Zealand) and Sodankyla in Finland (see Figure 1).

-> RAL against *in situ* profile

The smoothed XCH$_4$ *in situ* measurement $c_i$ is calculated as follows:

$$c_i = c_a + \boldsymbol{a_S}(\boldsymbol{x_i} - \boldsymbol{x_a}), \tag{4}$$

where $\boldsymbol{a_S}$ is the RAL column averaging kernel vector, $\boldsymbol{x_a}$ and $\boldsymbol{x_i}$ are the RAL *a priori* profile and *in situ* profile, respectively, $c_a$ is the RAL IASI *a priori* XCH$_4$.

For profile comparison, we also calculate the smoothed CH$_4$ profile *in situ* measurement $\boldsymbol{x_i}'$ as follows:

$$\boldsymbol{x_i}' = \boldsymbol{x_a} + \mathbf{A_R}(\boldsymbol{x_i} - \boldsymbol{x_a}), \tag{5}$$

using RAL's $\boldsymbol{A_R}$ averaging kernel matrix.

-> LMD against *in situ* profile

LMD IASI data only provides mtCH$_4$ together with the weighting function $\boldsymbol{w}$. There is no information about the *a priori* profile and the surface pressure.

$$c_i = \frac{\boldsymbol{w} \cdot \boldsymbol{dp} \cdot \boldsymbol{x_i}}{\sum \boldsymbol{w} \cdot \boldsymbol{dp}}, \tag{6}$$

### 3.4.2 Satellite vs FTIR measurements

When comparing the satellite and ground-based FTIR measurements, we need to take both the *a priori* profile and vertical sensitivity into account.

-> RAL against TCCON measurements

TCCON and RAL IASI data both provide their respective *a priori* profiles. Here, we use the TCCON *a priori* profile as the common *a priori* profile to adapt the RAL IASI data.

$$c'_{r,R} = c_{r,R} + c_{a,T} - c_{a,R} + \boldsymbol{a_S}(\boldsymbol{x_{a,R}} - \boldsymbol{x_{a,T}}), \tag{7}$$

where $c_{r,R}$ is the original RAL XCH$_4$ data, $\boldsymbol{a_S}$ is the RAL column averaging kernel vector, $\boldsymbol{x_{a,R}}$ and $\boldsymbol{x_{a,T}}$ are the RAL and TCCON *a priori* profiles, $c_{a,R}$ and $c_{a,T}$ are the RAL and TCCON *a priori* XCH$_4$, respectively. $c'_{r,R}$ thus corresponds with the RAL XCH$_4$, where its original *a priori* has been replaced by TCCON's *a priori* profile (Rodgers and Connor, 2003).

To take the vertical sensitivity of the RAL retrieval into account, we apply the smoothing correction on the retrieved FTIR profile. However, TCCON only delivers a total column-averaged mole fraction and no retrieved profile on which we could apply our sensitivity corrections. This is due to the fact that TCCON performs a scaling profile retrieval allowing for no variation in the profile shape. In this study, we calculate the ratio of the TCCON retrieved XCH$_4$ ($c_{r,T}$) to the *a priori* XCH$_4$ ($c_{a,T}$), and the ratio is then multiplied by the TCCON *a priori* profile $\boldsymbol{x_{a,T}}$ as the retrieved TCCON profile ($\boldsymbol{x_{r,T}}$). After that, we apply the smoothing correction using the RAL IASI column averaging kernel

$$c'_{r,T} = c_{a,T} + \boldsymbol{a_S}(\boldsymbol{x_{r,T}} - \boldsymbol{x_{a,T}}), \tag{8}$$

where $c'_{r,T}$ is the adapted TCCON XCH$_4$. The $\boldsymbol{x_{r,T}}$ is re-gridded to the RAL retrieval grid, so that the $c'_{r,T}$ and $c'_{r,R}$ have been computed on the same vertical layers. Here, we compare $c'_{r,T}$ with $c'_{r,R}$.

-> RAL against NDACC measurements

NDACC and RAL IASI data both provide the *a priori* profiles, and we apply the NDACC *a priori* profile as the common *a priori* profile to adapt the RAL IASI retrieved CH$_4$ profile

$$c''_{r,R} = c_{r,R} + c_{a,N} - c_{a,R} + \boldsymbol{a_S}(\boldsymbol{x_{a,R}} - \boldsymbol{x_{a,N}}), \tag{9}$$

where $c_{a,N}$ is the NDACC *a priori* XCH$_4$ and $\boldsymbol{x_{a,N}}$ is the NDACC *a priori* CH$_4$ profile. $c''_{r,R}$ thus corresponds with the RAL XCH$_4$, where its original *a priori* has been replaced by NDACC's *a priori* profile.

The retrieved NDACC CH$_4$ profile ($\boldsymbol{x_{r,N}}$) is smoothed with the RAL IASI column averaging kernel to consider the vertical sensitivity of the RAL IASI data.

$$c'_{r,N} = c_{a,N} + \boldsymbol{a_S}(\boldsymbol{x_{r,N}} - \boldsymbol{x_{a,N}}), \tag{10}$$

where $c'_{r,N}$ is the adapted NDACC XCH$_4$. The $\boldsymbol{x_{r,N}}$ is re-gridded to the RAL retrieval grid, so that the $c'_{r,N}$ and $c''_{r,R}$ have the same vertical ranges.

Here, we compare $c'_{r,N}$ with $c''_{r,R}$.

-> LMD against TCCON measurements

LMD IASI data only provides mtCH$_4$ together with the weighting function $\boldsymbol{w}$. LMD does not provide an *a priori* profile, so that it is not possible to apply *a priori* substitution as for RAL (see equations 7 and 9).

The LMD weighting function $\boldsymbol{w}$ is thus directly applied onto the scaled TCCON *a priori* profile $x_{r,T}$, which is used as a proxy for a TCCON retrieved profile. By doing this, we can not only include the vertical sensitivity of the LMD retrieval, but also reduce the uncertainty resulting from the TCCON near surface profile shape since the LMD weighting function is equal to 0 in the lower troposphere.

$$c''_{r,T} = \frac{\boldsymbol{w \cdot dp \cdot x_{r,T}}}{\sum \boldsymbol{w \cdot dp}}, \tag{11}$$

Here, we compare the LMD data $c''_{r,T}$ with $c_{r,L}$.

-> LMD against NDACC measurements

Similarly, we applied the LMD IASI weighting function $\boldsymbol{w}$ onto the retrieved NDACC CH$_4$ profile.

$$c''_{r,N} = \frac{\boldsymbol{w \cdot dp \cdot x_{r,N}}}{\sum \boldsymbol{w \cdot dp}}, \tag{12}$$

Here, we compare the LMD data $c''_{r,N}$ with $c_{r,L}$.

## 3.5 Measurement uncertainty

The uncertainty of each *in situ* profile is carefully estimated. For the vertical range within the *in situ* measurements, the uncertainty is from the reported measurements, with 1.3 ppb for IAGOS data (Filges et al., 2015), 1.5 ppb for AirCore data (Karion et al., 2010) and 1.5 ppb for HIPPO data (Wunch et al., 2010). For the vertical range above the *in situ* measurements, we use the difference between the model and the scaled model as the uncertainty. For the vertical range below the *in situ* measurements, the mean difference between the model and *in situ* measurement in the troposphere (below $\sim$150 hPa) is used as the uncertainty.

The combined uncertainty from satellite and *in situ* measurements is calculated as

$$\sigma_c = \sqrt{\sigma_{sat}^2 + \sigma_i^2}, \tag{13}$$

where $\sigma_{sat}$ is the uncertainty of satellite data, and $\sigma_i$ is the uncertainty of the *in situ* measurements. For the RAL measurement, the uncertainty is reported in the public data (about 35 ppb). For the LMD measurement, since there is no uncertainty value available, the stdv of the co-located satellite data is used as the uncertainty. Note that we only select the FTIR and satellite data pair, with more than 2 co-located satellite footprints.

### 3.6 Trend and seasonal variation

In this study, we derive the trend and seasonal variation of $CH_4$ between 1 July 2007 and 30 June 2015 (8 full years) from LMD and RAL measurements. We limit ourselves to this period to make the two satellite datasets have the same time coverage. According to the NOAA surface measurements (Dlugokencky et al., 1994), the global $CH_4$ mean concentration kept increasing

between July 2007 and June 2015, with an annual growth rate of 6.9±0.6 ppb/yr (WMO, 2017).

The level 2 satellite data are binned into $1° \times 1°$ grids to generate the level 3 daily means. The monthly data are created based on the daily data, and then the long-term trends and seasonal variations are calculated from the monthly means at each grid. To derive the trends from the month means $Y(t)$, with t the time in a fractional year, we use a regression model that includes a periodic function to describe the seasonal cycle:

$$Y(t) = A_0 + A_1 t + \sum_{k=1}^{3} (A_{2k} \cos(2k\pi t) + A_{2k+1} \sin(2k\pi t)), \tag{14}$$

where $t$ is in fraction of year, $A_0$ is the intercept, $A_1$ is the annual trend and $A_2$ to $A_7$ are the periodic amplitudes. Then, the de-trended data ($Y(t)_d$) is calculated as

$$Y(t)_d = Y(t) - (A_0 + A_1 \cdot t). \tag{15}$$

The seasonal variation is represented by the monthly means of the de-trended data and their associated uncertainty ($2\sigma$).

### 3.7 The discontinuity in RAL data after May $16^{th}$ 2013

Figure 3 shows the temporal evolution of all daily averaged data between 60°N and 60°S (left), and the prior- to post-day differences (right), for RAL total column (first row), RAL smoothed by the LMD sensitivity profile (second row), RAL's upper (6-12 km) partial column (third row), and RAL's lower (0-6 km) partial column ($4^{th}$ row). It clearly shows that, due to a change in the processing of the spectral response model on 16 May 2013, a 6.7±1.5 ppb discontinuity occurred in RAL's

retrieved total column methane (top). This issue has been reported in the RAL product user guide (https://catalogue.ceda.ac. uk/uuid/f717a8ea622f495397f4e76f777349d1; last access: 12 January 2023). Note that no such effect is visible in the LMD data (not shown) nor can we clearly distinguish a discontinuity in RAL's mid-troposheric $mtCH_4$ concentrations, obtained by smoothing the RAL profiles with the LMD sensitivity profile, from the overall variability in the data (Figure 3 second row). It is found that this issue effects the RAL's lower partial columns (0-6 km by ~9.6±2.2 ppb) to a greater extent than higher

layers (6-12 km by ~3.6±1.1 ppb), which might explain the more limited impact on the LMD smoothed RAL profiles. The above values were determined by taking, for each day, the median $CH_4$ concentration between 60°N and 60°S. From this we calculated the difference between the median concentration value prior and after the day in question and determined the value that corresponded with the May $16^{th}$ 2013 transition (the difference between the median concentration on the $17^{th}$ and $15^{th}$ of May). In all cases, apart from RAL's LMD smoothed $mtCH_4$ (0.2±2.7 ppb), this is the most prominent feature

in the day-to-day variability plot (Figure 3 (right)). As an indicator of the uncertainty, we took the standard deviation of all these day-to-day difference values 1.5 months prior and after the transition. Note that we take on a single correction factor

only, without a latitudinal or seasonal dependency. This was investigated and differences do appear but when taking their (considerable) uncertainties into account, none of the data subset correction factors showed a deviation from the general 60°N-60°S correction, described above, that was statistically significant.

As such this issue complicates our analysis and, depending on the quality parameters we wanted to explore, we have either focused on a particular year, applied a simple +6.7 ppb correction on RAL's post May 16$^{th}$ 2013 XCH$_4$ total column concentrations (+9.6 and +3.6 ppb in case of RAL qCH$_4$ partial columns) or have regarded the pre- and post-May 16$^{th}$ 2013 RAL measurements as 2 independent datasets after which the quality parameters are averaged, using the covered time frames as weights. The latter method has the advantage of not having to add a correction parameter which adds additional uncertainty. On the downside, cutting the timeseries in two, leaves us with a relatively short 3 year 2013-2016 timeperiod, resulting in significantly more uncertainty on the obtained statistical parameters when the data density is low. Therefore, unfortunately, since regarding the RAL data as two independent datasets may often be considered as the best solution, the limited data density of reference measurements at many sites, and the fact that the then obtained parameters would greatly depend on the temporal range of the reference measurement prompts us to primarily use a post May 16$^{th}$ bias shift. We have indicated in each case what (if any) correction method has been used. The potential impact of each of these correction methods is further discussed in Section 5.2.

## 4    Product analysis

In this section, prior to our RAL-LMD intercomparisons and validation with reference data, we looked at several parameters within each of the datasets. In particular we were interested in the internal consistency of day-night, scan angle, residual cloud cover and IFOV to IFOV differences. The latter is done for RAL only as this information is not present in the LMD product which takes the average of the four IFOV within a FOR. This was done by drawing up histogram plots and looking at the distribution of the global data (not shown here) for the 2014 October month. We also specifically looked at RAL's partial columns differences.

### 4.1    Internal consistency

For mtCH$_4$ LMD we observed very small day-night differences in the distribution over land with a slightly lower mean (~4 ppb) for daytime data compared to night-time data. Also, mtCH$_4$ values are slightly higher (~7 ppb) for the edge viewing angles than for the nadir measurements.

For XCH$_4$ RAL we observe slight day-night differences (within 5 ppb), in the averaged distribution. The day uncertainties (typical stdv of ~15-20 ppb) are, as expected, lower than night uncertainties (typical stdv of ~35-40 ppb). Also its nadir values are higher (up to 11 ppb) than its edge viewing angle data on the monthly and global mean XCH$_4$ especially over sea. Also, the nadir measurements exhibit lower retrieval uncertainties (~5 ppb over land/day on the median of the global distributions). Concerning the inter-IFOV differences, the highest differences are observed for daytime XCH$_4$ and between IFOV 3 and IFOV 1 with averaged differences of about 7 ppb over land and 8 ppb over sea. The inter-IFOV retrieval uncertainties are all within

ppb. This difference is unexpectedly large and should be further investigated at the L1 data processing level. However, since these kind of inter-IFOV analysis that focuses on radiometric biases are not available in the IASI public reports, we were not able to derive a clear instrumental effect explaining the IFOV 1-IFOV 3 relative departure. Also, filtering with IASI L2 cloud fraction had a slight impact (about 3 ppb on average) on the global distribution for the XCH$_4$. There is a slight decrease of the retrieval errors of about 2 ppb on averaged for cloud fraction <15%. All this indicates that the RAL cloud filtering condition already eliminates most of could affected scenes.

The above analysis does not exclude stronger differences on a regional scale. For instance, strong negative day-night (with higher nighttime values) differences can be observed over desert regions (see Figure 4) in both LMD and RAL. Surface emissivity is difficult to handle in some areas of the Sahara where it is particularly low. This typically causes a negative difference, which is larger in day than night due to the high surface-air temperature contrast. Likewise high surface-air temperature contrasts can trigger the elevation of surface emissions and can thus induce a positive day-night difference. Note that all biases are present with differences in seasonal variations and in various regions, and spectral and angular dependencies vary between different land types and surface topologies, respectively, in different areas.

## 4.2 RAL partial columns

The DOFs of RAL indicate that, apart from the higher latitudes (>60° North and South), 2 independent partial columns can be obtained from the retrieved profiles. Therefore, in this section, we calculate for both RAL and the smoothed CAMS profiles the monthly averaged partial columns between 0-6 km (lower layer) and 6-12 km (upper layer) for all years. In Figure 5 we show 2012 as an example year to compare RAL with the CAMS model. Small inter-annual absolute value differences do occur, but the observations and conclusions discussed below remain the same. We also need to point out that the RAL product comes with a 50 layer column averaging kernel, but the profile averaging kernel is a 5x50 matrix where the smallest dimension corresponds with the lowest 5 levels of a coarser 12 level retrieval pressure grid. The 3 lowest levels of this lower resolution grid correspond with 1000 hPa, 422 hPa and 178 hPa respectively. The latter 2 pressure levels correspond with the limits of the 0-6 and 6-12 km altitude range of the partial columns. While these pressure ranges roughly contain 1 DOFs each, one cannot specifically select, due to the low-resolution grid, the partial column vertical range based on the DOFs for each measurement and therefore we cannot state that these column layers are fully independent in all cases.

Figure 5 shows the differences between RAL and CAMS qCH$_4$ values in the upper and lower layers in January, April, July and October 2012. The mean and stdv of the differences are only calculated for the low- and mid-latitude regions (<60° North and South). It is apparent that the qCH$_4$ observed by RAL is generally 3.1-8.0 ppb larger than the CAMS model in the upper layer and 5.5-7.7 ppb lower than the CAMS model in the lower layer. Specifically, the RAL qCH$_4$ in the lower layer is generally lower than the CAMS model in the Mediterranean area, tropics, east Asia and south America, depending on the month of year. The mean underestimation in the Pacific Ocean between 15°N and 15°S during these 4 months is 12.5 ppb less than the CAMS model.

Based on the stdv of the differences, the spatial variability between the RAL and CAMS qCH$_4$ in the upper layer is less than that in the lower layer. The difference of qCH$_4$ between the upper and lower layers from RAL and CAMS are also shown in

Figure 5. For RAL the mean upper-lower difference ranges between 6.3 and 16.5 ppb, while for CAMS the difference ranges between -5.7 and 7.3 ppb. For CAMS, in most conditions, the difference between upper and lower qCH$_4$ is either very small or the lower layer yields higher concentrations than the upper layer. A notable exception is the band of positive (upper-lower) bias values located around the Southern hemisphere sub-tropics, which is more pronounced in summer than in winter. This

latitudinal structure is equally captured by RAL, but the difference between upper and lower qCH$_4$ is far more pronounced in a far wider region throughout the whole year, also peaking in summer and autumn. Note that none of these features are inherent to the RAL *a priori*, which exhibits a uniform near 0 partial column bias apart from the polar regions where the lower partial column is ~25 ppb higher than the upper partial column, and this probably stems from the lack of sufficient spectral information.

Also apparent is the often stark contrast between adjacent land and sea measurements. Some striking examples of this situation are Australia in October and Northern Europe in April. These features are not replicated in the CAMS partial column biases, which shows (as expected) a smooth transition from land to sea, even though the relevant averaging kernel smoothing has been applied. While we expect differences in sensitivity to occur between land and sea measurements (a change in the retrieval uncertainty and with that the DOFs is expected), ideally the impact thereof is translated into the averaging kernel.

Note that these features are not clearly present in the LMD and RAL total column product.

## 4.3    Short summary

While most parameters investigated point to no major issues (i.e. day-night diffences over the sahara desert can be readily explained), RAL's inter-IFOV bias prompts further investigation. The upper-lower qCH$_4$ partial column difference is consistent with that observed by CAMS but here again the difference between adjacent land and sea partial column differences requires

further investigation. Other points where RAL differs significantly with CAMS the far more pronounced (in magnitude and time) band of positive (upper-lower) bias values located around the Southern hemisphere sub-tropics.

## 5    Direct intercomparison

In order to directly compare RAL with LMD retrievals we need to consider some inherent differences between the satellite products first. Foremost, and already discussed, are the differing sensitivities as a function of altitude. Another source of

differences is that RAL selects a single IFOV with the warmest brightness temperature among the 4 of them within any given IASI FOR, whereas LMD uses a combination thereof, so a direct comparison on a measurement by measurement basis is impossible. Instead we opted to use the CAMS model as an intermediate. Not only can we compare the gridded satellite products to the model and to one another, but we can also compare their respective biases towards the model. Doing so overcomes to a great extent the fact that, even when looking at the bias between LMD and RAL_LMDavk mtCH$_4$, differences

in vertical sensitivity remain. We should also point out at this stage that the model data is no substitute for reality and that it can harbour errors of its own. Of particular concern, particularly with respect to comparisons with the LMD data as it is more sensitive at higher altitudes, is the accuracy of the location of the UTLS transition zone within the model.

## 5.1 Absolute differences

Figure 6 shows the monthly mean global bias (for January, April, July and October 2012) between the satellite products and the CAMS model, whereby the model is always smoothed with the respective sensitivity/averaging kernel profile. This is done for both LMD (top row), RAL (second row) and RAL smoothed by the LMD sensitivity profile (third row). As one can see all products have their distinct regional and seasonal biases with respect to CAMS. All products seem to feature stronger biases at high latitudes, with LMD featuring particularly strong negative biases around the month of October in the Northern Boreal regions and RAL (total column and LMD smoothed) featuring strong positive values compared to CAMS (particularly over Northern latitudes in April and over Antarctica in January, RAL inland Greenland being a curious exemption to this pattern). To limit the impact of these regions, the overall monthly mean biases as shown in the figure are drawn up from all values within 60° North and South, in line with LMD's recommended latitude range. We also found a few cases in which the application of the RAL column averaging kernel onto the CAMS profile yielded clear erroneous outliers. These have been filtered out using a interquartile-distance filter. No more that 5 measurements needed to be removed for each month. Looking at the thus obtained values we see that the RAL column averaged product features the lowest bias with respect to CAMS, and with lower scatter than LMD. The overall bias between RAL_LMDavk and CAMS on the other hand is very similar to that of LMD-CAMS. Its scatter (stdv of the RAL_LMDavk-CAMS differences) is similar to that of total column RAL.

The bottom two rows in Figure 6 feature the comparison between LMD and RAL_LMDavk ($4^{th}$ row) and finally the difference in the respective biases of LMD and RAL_LMDavk with respect to CAMS (bottom row). This last comparison should, in theory, have minimized most of the residual sensitivity differences between both products and is thus the most accurate representation of their respective overall differences. The direct comparison between LMD and RAL_LMDavk ($4^{th}$ row) still yields overall negative bias values in excess of -10 ppb. This disappears to a large extent when looking at their respective biases towards CAMS (bottom row), indicating that, even when smoothing RAL with the LMD sensitivity profile, their inherent sensitivity differences remain substantial. This observation is important when interpreting further comparison results. Also, while the average bias is small, we can still observe significant regional and seasonal biases between the products. To highlight just a few areas, in January we observe large positive LMD-RAL bias values over the Pacific between 10°N and 30°N as well as more moderate positive biases over the entire Northern Hemisphere Atlantic Ocean and Western Europe. In October, this positive bias band has shifted towards the Southern hemisphere, forming a positive latitudinal bias belt between 10°S and 30°S over land and sea. Strong negative biases are observed over the Canadian Boreal forests. The latter biases disappear in April, while the positive biases over the ocean become less outspoken. Strong positive biases are now observed over Eastern Europe. In July, the previous 20°N oceanic positive bias belt relocates to the Southern Hemisphere, while over land strong positive biases are observed in Northern Egypt, East of the Caspian Sea, and the Central and Eastern United States. Strong negative biases occur over Indonesia and the Northern Pacific, although note that the most significant biases occur at >60°N, outside the LMD domain.

Figure 7 shows (as the bottom row in Figure 6) the differences in the respective biases of LMD and RAL_LMDavk with respect to CAMS for different months (January, April, July and October), but now for several years (2008, 2010, 2012, 2014).

One immediate observation is that the average LMD-RAL difference in their respective biases towards CAMS, becomes ever more positive when moving from 2008 to 2014. This is consistently seen for all months. We did not apply any correction to the 2014 RAL data so its discontinuity could be at play, but the trend is also clearly visible when moving from 2008 to 2012. This points to a temporal stability issue with either LMD, RAL or both. The magnitude of these overall averaged bias shifts amount to a 2.3 ppb/yr (January), 2.1 ppb/yr (April), 1.1 ppb/yr (July), 1 ppb/yr (October) shift. The marked differences between January-April on the one hand and July-October on the other hand also alludes to a seasonal error component.

While we do see shifts in the magnitude of some features (for instance in January and October we clearly see ever stronger positive biases over Australia), no major shifts in the overall patterns are observed. For instance, all years still feature strong negative biases over the Canadian Boreal forests in January and October, and all years show the positive bias band (in January positioned between 10°N and 30°N, in October between 10°S and 30°S). A new feature that can be clearly observed in the 2014 (last row) data is the emergence of a, somewhat weaker but still clearly positive, second bias band in the opposite hemispheres (in January positioned between 10°S and 30°S, in October between 10°N and 30°N). The emergence of this second band is also already apparent in 2012.

## 5.2 Long-term trend and seasonal variation

The observation of clear changes in the biases as a function of time in Figure 7 prompts the exploration of the long-term trends and seasonal variations of both LMD and RAL. The $CH_4$ annual growths derived from LMD and uncorrected RAL (thus for now ignoring the discontinuity) are compared to each other (Figure 8). Due to the cloud contamination and post-filtering, the $CH_4$ measurements are not always available even though we use the monthly averaged data. In this section, we only consider the trend on the 1x1° grid where there are less than 12 absent monthly means during these 8 years. The $mtCH_4$ trends derived from the LMD data are generally available in the low-latitude regions, while the $XCH_4$ trends derived from the RAL are calculated in most places except for the polar region. The mean and stdv of the $CH_4$ annual growth rates are 6.43±1.34 ppb/yr and 4.06±0.66 ppb/yr derived from the LMD and RAL data, respectively. The mean difference in $CH_4$ trend from LMD and RAL measurements is 2.54 ppb/yr, which is larger than the stdv of their differences of 1.53 ppb/yr. After smoothing the RAL data with LMD weighting function (RAL_LMDavk), the $mtCH_4$ trend derived from the RAL_LMDavk is 4.29±0.86 ppb/yr, which is larger than the $XCH_4$ trend derived from the original RAL data. The mean difference in $mtCH_4$ trend between LMD and RAL_LMDavk reduces to 1.63 ppb/yr, but is still larger than the stdv of their differences (1.33 ppb/yr).

The global maps of $CH_4$ annual growth rates are also derived from LMD and RAL night-time measurements (not shown here). The spatial distributions of $CH_4$ trend derived from the daytime and night-time measurements are similar for both LMD and RAL. Moreover, the global mean and stdv of the $CH_4$ trend derived from night-time measurements are 6.65 and 1.46 ppb/yr derived from the LMD data, and are 4.01 and 0.68 ppb/yr derived from the RAL data, which are close to the results derived from daytime measurements. As the results from daytime and night-time measurements are consistent, we only discuss the trends of $CH_4$ derived from the RAL and LMD daytime measurements in the following sections.

The time series and seasonal variation of $CH_4$ are further investigated based on the TransCom (Figure 9)(Gurney et al., 2002), which has been used in the Carbon Tracker-$CH_4$ model (Bruhwiler et al., 2013), including 11 land (Figure 10) and

11 ocean (Figure 11) regions. A 6.7 ppb post May $16^{th}$ 2023 correction has been applied to the RAL total column data. Here however, we mainly focus on LMD and RAL_LMDavk $mtCH_4$ measurements. At land regions, it is found that the seasonal variations of $mtCH_4$ from LMD and RAL_LMDavk measurements are generally close to each other in the low-latitude regions, but are different in the high-latitude regions. Specifically, the seasonal variations of $mtCH_4$ from LMD and

RAL_LMDavk measurements are close to each other in South American Tropical, South American Temperate, Northern Africa, Eurasia Temperate and Tropical Asia, while they are different at North American Boreal, North American Temperate, Europe, Southern Africa, Eurasia Boreal and Australia. The $mtCH_4$ annual growths derived from LMD are 0.4-1.8 ppb/yr larger than RAL_LMDavk in most regions except at North American Boreal and Eurasia Boreal. The $mtCH_4$ annual growth derived from LMD has a strong latitude dependence, which is close to 6 ppb/yr in the tropical region but less than 3 ppb/yr in

the high-latitude regions. At ocean regions, it is found that the seasonal variations of $mtCH_4$ from LMD and RAL_LMDavk measurements are close to each other in most regions, except for the Northern Ocean and North Atlantic Temperate. At the Southern Ocean and South Pacific Temperate, the phases of the seasonal variations of $mtCH_4$ from LMD and RAL_LMDavk measurements are similar, but the amplitudes of the seasonal variation of $mtCH_4$ derived from LMD measurements are larger than those derived from the RAL_LMDavk data. The $mtCH_4$ annual growths derived from LMD are 0.3-2.2 ppb/yr larger than

RAL_LMDavk in most ocean regions except at the Southern Ocean.

The above analysis shows that the annual growth of $mtCH_4$ derived from the uncorrected RAL data between July 2007 and June 2015, while generally consistent between regions, is systematically smaller than that of LMD. While we could not clearly extract a correction factor for the May $16^{th}$ 2013 RAL discontinuity with regards to the LMD smoothed RAL_LMDavk $mtCH_4$ values (see Section 3.7), the here observed discrepancies nevertheless prompt us to produce trend estimates based on 2

individual linear trends for the two periods before and after 16 May 2013. We then average, using the covered time frames as weights, the two estimates to get an overall corrected value for the trend.

Table 3 lists all the trends of $CH_4$ between July 2007 and June 2015 derived from LMD, RAL_LMDavk and RAL_LMDavk (2 periods) at 9 TransCom low-and mid-latitude land regions. The weighted mean of the annual growths of $XCH_4$ becomes 5.6 ppb/yr by using the RAL data before and after 16 May 2013. This result is close to the $mtCH_4$ annual growth of 5.3 ppb/yr

between July 2007 and June 2015 observed by the LMD data. The mean $XCH_4$ annual growth derived from the RAL data between July 2013 and June 2015 is 9.5 ppb/yr, which is larger than that of 4.4 ppb/yr between July 2007 and May 2013. The $CH_4$ annual growth rate derived from the NOAA surface *in situ* measurements between June 2013 and June 2015 is 11.2 ppb/yr, which is also larger than that of 5.6 ppb/yr between July 2007 and May 2013.

To further explore the observed RAL and LMD differences as well as the impact (if any) of the discontinuity corrections,

Figure 12 shows the long term trend (first three figures) and seasonal cycle amplitudes (bottom figure) of the respective LMD, RAL and RAL_LMDavk Satellite product-CAMS biases, grouped per $10°$ latitude band. The top row shows the behaviour of all data as is, the second row when applying a +6.7 ppb correction onto the post May $16^{th}$ 2013 RAL total column data and the third row, when splitting the RAL total column and RAL_LMDavk into 2 independent timeseries. Uncorrected (top figure), the overall trend of the RAL_LMDavk-CAMS bias shows no significant latitudinal dependence, while RAL features higher values

at high latitudes. When applying a 6.7 ppb correction (second figure) the RAL values shift upwards (less negative values) by

**Table 3.** The trend of $CH_4$ in unit of ppb/yr between July 2007 and June 2015 derived from LMD, RAL_LMDavk and RAL_LMDavk (2 periods) at 9 TransCom low- and mid-latitude land regions.

| Region | LMD | RAL_LMDavk | RAL_LMDavk (2 periods) | |
|---|---|---|---|---|
| | | | 2007.7-2013.5 | 2013.6-2015.6 |
| North American Temperate | 4.8±0.8 | 4.2±0.7 | 4.1±1.5 | 9.3±4.1 |
| South American Tropical | 5.0±1.1 | 3.8±0.7 | 4.2±1.2 | 10.0±6.6 |
| South American Temperate | 5.2±0.7 | 4.4±0.5 | 4.0±0.5 | 12.4±5.5 |
| Europe | 4.0±1.0 | 3.6±0.7 | 3.8±1.4 | 6.2±2.1 |
| Northern Africa | 5.5±1.2 | 4.5±0.7 | 4.5±1.5 | 9.1±1.8 |
| Southern Africa | 6.3±0.8 | 4.5±0.9 | 4.5±1.1 | 11.8±7.8 |
| Eurasia Temperate | 4.9±1.4 | 4.1±0.9 | 4.3±1.6 | 7.5±4.8 |
| Tropical Asia | 5.7±1.3 | 4.4±0.8 | 5.3±1.3 | 9.6±5.5 |
| Australia | 6.0±0.6 | 4.5±0.5 | 4.5±0.9 | 9.4±1.9 |
| mean | 5.3 | 4.2 | 4.4 | 9.5 |
| | | | 5.6 | |

approximately 1 ppb. When splitting the RAL-CAMS and RAL_LMDavk-CAMS data into 2 independent timeseries (third figure), we see the strongest impact, with the RAL bias shifting further upwards by another ~1 ppb. For RAL_LMDavk, not only a bias shift is observed but also the shape has changed considerably (from a constant offset to one which shows much weaker negative biases near the poles). It appears that the impact of the latter correction is much stronger near the poles (~3

5    ppb) than at the (sub)tropics (~1 ppb). Implementing a stronger (in stead of 6.7 ppb, a 10 ppb shift) correction into both RAL timeseries does bring the outcome of the 2 correction methods closer to one another, however the change in the shape of RAL_LMDavk (using 2 independent time series) with respect to its latitudinal dependence of the long term trend could not be replicated with a simple bias correction. Moreover such a significant change would certainly have been picked up in our analysis (See Section 3.7).

10    Using the last correction method (2 independent time series) all three algorithms show similar trend values near the (sub)tropics between 30°S and 30°N. Further North and South however LMD-CAMS shows markedly ever stronger negative trend values when moving towards the poles. We see little to no impact of the correction methods onto the observed seasonal cycle amplitudes of the respective Satellite-CAMS bias (bottom figure). It shows that the amplitude of the seasonal cycle in the RAL-CAMS and RAL_LMDavk-CAMS residuals are consistently lower than for LMD-CAMS. This points to a significant

15    difference between the seasonal cycle phases or amplitudes of both products. Also of interest is the observation of a strong increase in the LMD-CAMS residual seasonal amplitudes at higher (>50°) latitudes in line with LMD's very strong decrease in the residual trend at latitudes exceeding (40°) in both hemispheres (Figure 12 bottom).

### 5.3 Short summary

Our analysis of the direct comparisons between RAL and LMD paint a rather complex picture with observed marked differences both in space and time. Important to note is that even when smoothing RAL with the LMD sensitivity profile, vertical sensitivity differences with LMD remain as is shown by the difference when comparing LMD and RAL_LMDavk directly or their respective biases towards CAMs. Adding further complexity is the impact of RAL's May $16^{th}$ 2013 discontinuity, which depending on the correction method used, impacts the long term trend by 1 to 2 ppb/year. Some of the most marked RAL-LMD differences observed, point to a significant shift in the amplitude and/or phase of their respective seasonal cycles, particularly at higher latitudes. North American Temperate, Europe and Northern Ocean in Figures 10 and 11 are prime examples. Also at higher latitudes, we see ever stronger differences in the long-term trends of both products.

## 6 Comparisons with independent reference data

In this section we compare RAL (discontinuity corrected) and LMD data with *in situ* data from HIPPO, IAGOS and AirCore as well as ground-based remote sensing data from the TCCON and NDACC networks. Note that given the complex nature of the RAL-LMD differences (both in time and in space), any obtained difference observed between the satellite and reference measurements depends on the time and location of the reference measurements in question. For instance, HIPPO measurements focused largely on the Pacific Ocean and often measured near the poles. IAGOS profiles are taken during ascent and descent from/towards airports and are thus tilted towards more urban environments. AirCore is restricted to just a few location in the United States, New Zealand, and Finland (see Figure 1). There are also large differences with respect to the time periods covered.

### 6.1 Comparisons with *in situ* profiles

In this section, LMD mtCH$_4$ and RAL XCH$_4$ are compared to the *in situ* profiles. We also look at RAL's 0-6 km and 6-12 km qCH$_4$ partial columns. The latter is possible as the degrees of freedom (DOFs) of the RAL CH$_4$ profile is about 2.0, with two distinct pieces of information in partial columns of 0-6 km and 6-12 km.

#### 6.1.1 RAL Total column

The RAL and HIPPO XCH$_4$ together with their differences along with the latitude are shown in Figure 13. Both RAL and HIPPO measurements observe high XCH$_4$ in the Northern Hemisphere and low XCH$_4$ in the Southern Hemisphere. Specifically, the XCH$_4$ at 40°N is about 80 ppb larger than that at 40°S. Two XCH$_4$ peaks at about 35°N and 75°N are captured by both datasets. Only 49 out of 466 (10.5%) differences between RAL and HIPPO measurements are outside their combined 1 $\sigma$ uncertainties. However, the mean of HIPPO measurements is 16.5 ppb larger than the mean of RAL measurements between 15°N and 15°S, with many differences beyond the combined uncertainties. The overall mean and stdv of the differences between RAL and HIPPO measurements are -4.6 ppb and 16.5 ppb, respectively. The scatter plot between RAL and HIPPO

measurements shows that the correlation efficiency (R) is 0.84, indicating there is a good agreement between RAL and HIPPO measurements. The linear fit suggests that the RAL data is slightly less/greater than the HIPPO measurements when the $XCH_4$ is low/high. Note that since all HIPPO measurements occurred prior to May $16^{th}$ 2013, no correction method needed to be applied.

Furthermore, the RAL $XCH_4$ are also compared to IAGOS and AirCore (not shown here). Here we did apply a +6.7 ppb correction on the post-May $16^{th}$ 2013 data. The mean and stdv of the differences between RAL and IAGOS measurements are -1.6 ppb and 21.9 ppb respectively (note that without correction the bias equaled -4.8 $\pm$ 23.0 ppb ). The R between the RAL and IAGOS measurements is 0.52. 222 out of 260 differences between RAL and IAGOS measurements are within their combined 1 $\sigma$ uncertainties. The mean and stdv of the differences between RAL and AirCore measurements are -4.4 ppb and

14.1 ppb, respectively (uncorrected the bias equals -10.2 $\pm$ 14.5 ppb). The R between RAL and AirCore measurements is 0.83 and the linear fit is close to the one-by-one line, indicating there is a good agreement between RAL and AirCore measurements. Indeed 45 out of 49 differences between RAL and AirCore measurements are within their combined uncertainties. Of the 3 *in situ* reference datasets, IAGOS typically features the lowest correlation efficiency (R) and highest stdv of the differences. This is due to a combination of having a far greater distribution around the globe and having profiles that are taken at or near urban

centers (and thus local emission sources) in stead of remote locations. Its profiles also typically require more extrapolation.

### 6.1.2   LMD mid-tropospheric column

Figure 14 shows the LMD and HIPPO $mtCH_4$ together with their differences along with the latitude. The data density of the LMD data is much less than the RAL data, but co-located LMD measurements are still able to observe the high $mtCH_4$ in the Northern Hemisphere and the low $mtCH_4$ in the Southern Hemisphere as expected. LMD nicely captures the overall

latitudinal distribution of $CH_4$ with no obvious issues. As already mentioned in Section 3.4, the stdv of the co-located LMD measurements are calculated as the retrieval uncertainty of the LMD data because of no reported uncertainty. As a result, 34 out of 97 differences between LMD and HIPPO measurements are within their combined uncertainties. The mean and stdv of the differences between LMD and HIPPO measurements are -10.9 ppb and 27.7 ppb, respectively. The R between LMD and HIPPO measurements is 0.48.

Similarly, the IAGOS and AirCore measurements are used to compare with co-located LMD data. The mean and stdv of the differences between LMD and IAGOS measurements are 2.3 ppb and 30.6 ppb, respectively. The R between LMD and IAGOS measurements is 0.49. Only 27 out of 58 differences between LMD and IAGOS measurements are within their combined uncertainties. Only 3 co-located LMD and AirCore are selected, and 2 out of 3 differences between LMD and AirCore measurements are within their combined uncertainties. The mean and stdv of the differences between LMD and

AirCore measurements are -0.3 ppb and 15.2 ppb, respectively. The R between LMD and AirCore measurements is 0.60.

    The mean and stdv of the differences, together with the R and N (the number of measurement pairs) are summarized in Table 4.

**Table 4.** The mean and stdv of the difference between *in situ* and IASI satellite $CH_4$ measurements (RAL measurements feature a discontinuity correction).

| *in situ* | IAGOS | | AirCore | | HIPPO | |
|---|---|---|---|---|---|---|
| Satellite | LMD | RAL | LMD | RAL | LMD | RAL |
| mean (SAT-AIR) [ppb] | 2.3 | -1.6 | -0.3 | -4.4 | 10.9 | -4.6 |
| stdv (SAT-AIR) [ppb] | 30.6 | 21.9 | 15.2 | 14.1 | 27.7 | 16.5 |
| R | 0.49 | 0.52 | 0.60 | 0.83 | 0.48 | 0.84 |
| N | 58 | 260 | 3 | 49 | 97 | 466 |

#### 6.1.3 RAL Partial columns

Figure 15 shows the RAL and HIPPO $qCH_4$ together with their differences along with the latitude in the vertical ranges of 0-6 km and 6-12 km. Again we have to note that the HIPPO vertical profiles have been expanded with scaled CAMS model data. The mean and stdv of the differences between RAL and HIPPO measurements in the 0-6 km layer are -12.2 ppb and 26.5 ppb, respectively. The mean and stdv of the differences between RAL and HIPPO measurements in the 6-12 km layer are -1.6 ppb and 22.2 ppb, respectively. The R between RAL and HIPPO measurements are 0.87 and 0.76 in the 0-6 km and 6-12 km layers. The 0-6 km partial column (Figure 15 (top)) shows a consistent $qCH_4$ upward trend with latitude in the Northern Hemisphere. For the 6-12 km partial column (Figure 15 (bottom)), two $qCH_4$ concentration peaks can be observed around 35°N and 75°N. The HIPPO measurements are larger than the RAL data in 0-6 km layer. For this layer 325 out of 466 differences between RAL and HIPPO measurements are within their combined uncertainties, and the underestimation of RAL measurements is particularly found in the tropical region. For the 6-12 km layer, 343 out of 466 differences between RAL and HIPPO measurements are within their combined uncertainties, and the RAL and HIPPO measurements are close to each other in the tropical region. The stdv of the differences between RAL and HIPPO measurements in both partial columns are larger than that in the total column, reflecting that the uncertainties of the partial columns (0-6 km and 6-12 km) are larger than that of the total column.

The mean and stdv of the differences between RAL measurements and IAGOS (expanded with scaled CAMS model data) in the 0-6 km layer are -5.0 ppb and 35.1 ppb, respectively. The mean and stdv of the differences between RAL and measurements in the 6-12 km layer are -3.2 ppb and 22.7 ppb, respectively (without the discontinuity correction -Here +9.6 ppb for the 0-6 km layer, and +3.6 ppb for the 6-12 km layer respectively- the biases were -9.5±36.0 ppb and -4.9±23.2 ppb for the 0-6 km and 6-12 km layer respectively). The R between RAL and IAGOS measurements are 0.60 and 0.54 in the 0-6 km and 6-12 km layers. For the lower layer (0-6 km), 179 out of 260 differences between RAL and IAGOS measurements are within their combined uncertainties. For the upper layer (6-12 km), 158 out of 260 differences between RAL and IAGOS measurements are within their combined uncertainties. The mean and stdv of the differences between RAL and AirCore measurements in 0-6 km are -14.3 ppb and 26.3 ppb (uncorrected -22.5 ppb and 27.1 ppb), respectively. The mean and stdv of the differences between RAL and AirCore measurements in the 6-12 km layer are -7.5 ppb and 26.1 ppb (uncorrected -10.6 ppb and 26.1 ppb),

respectively. The R are 0.77 and 0.52 in the 0-6 km and 6-12 km layers. For the lower layer (0-6 km), 37 out of 49 differences between RAL and AirCore measurements are within their combined uncertainties in 0-6 km. Only 22 out of 49 differences between RAL and AirCore measurements are within their combined uncertainties in 6-12 km. The standard deviation and R results for IAGOS are again markedly worse than for HIPPO and AirCore.

5    A summary of the RAL patial column comparison results with *in situ* profile measurements are listed in Table 5.

**Table 5.** The mean and stdv of the difference between IASI RAL partial columns (a bias correction has been implemented) and *in situ*.

| *in situ* | IAGOS | | AirCore | | HIPPO | |
|---|---|---|---|---|---|---|
| RAL | 0-6 km | 6-12 km | 0-6 km | 6-12 km | 0-6 km | 6-12 km |
| mean (SAT-AIR) [ppb] | -5.0 | -3.2 | -14.3 | -7.5 | -12.2 | -1.6 |
| stdv (SAT-AIR) [ppb] | 35.1 | 22.7 | 26.3 | 26.1 | 26.5 | 22.2 |
| R | 0.60 | 0.54 | 0.77 | 0.52 | 0.87 | 0.76 |
| N | 260 | 260 | 49 | 49 | 466 | 466 |

One striking feature, observable in the 0-6 km bias as a function of latitude plot (Figure 15 (top,left)), is a marked negative bias with respect to HIPPO near the equator. This corresponds with our observations using the CAMS model (Figure 5 (second row)), where a narrow band of negative RAL-CAMS biases can be seen over the Pacific Ocean, near the equator in nearly all seasons.

## 10  6.2   Comparisons with ground-based FTIR measurements

Here we compared the LMD and RAL methane data products with ground-based remote sensing data from the TCCON and NDACC networks. As with the *in situ* comparisons, a +6.7 ppb correction has been applied to the RAL post May 16[th] 2013 data. Also note that there is substantial difference in the time-periods covered by the individual stations. For TCCON, the stations that cover almost the entire time period (less than 2.5 years of missing data), and hence-forward referred to as core

15  stations, are: Sodankyla, Bialystock, Orleans, Garmish, Park Falls, Lamont, Izana, Darwin, Wollongong and Lauder. Other stations on the other hand have noticeably shorter coverages (Rikubetsu and Edwards for instance have less than 2 years of co-located measurements). For NDACC, all stations are listed as core stations as they cover quasi the entire timeperiod, apart from Maïdo (2.5 years of data), which is excluded. Note that both Mauna Loa and St-Denis feature some large (> 1 year) data gaps in their time series and that even with a long time span, the number of co-located data pairs may differ greatly between

20  stations. For instance, at high latitude sites (Eureka, Thule) annual gaps occur in the dataset during wintertime (see Figures 16 and 17). For RAL, the amount of pre- versus post-discontinuity data largely determines the magnitude of the impact of the applied discontinuity-correction and therefore when comparing average overall long-term trends we have restricted ourselves to the so-called core stations which cover a substantially long time period as listed above.

### 6.2.1 RAL bias and scatter

As the ground-based FTIR measurements (both TCCON and NDACC) have limited vertical information in the troposphere, we only focus on the total column of RAL in this section. Figure 16 (left) shows the time series (May $16^{th}$ 2013 uncorrected) of the differences between the RAL IASI and TCCON 2-weekly means at 21 sites between July 2007 and June 2015. The sites are sorted by their latitudes from North to South. The absolute mean and stdv of the, +6.7 ppb corrected, differences (RAL-TCCON) at all sites are 5.05 and 11.23 ppb (uncorrected this equated to 4.31 ppb and 11.28 ppb), respectively. The differences are within $\pm$ 20 ppb, and there is no clear seasonal variation in the differences at most sites. For high-latitude sites (Eureka and Sodankylä), the RAL $XCH_4$ is larger than the TCCON measurements, especially in spring. Ostler et al. (2014b) pointed out that the smoothing error of TCCON $XCH_4$ retrieval is large under the polar vortex situation, and the TCCON measurement is about 40 ppb larger than the real status. However, we find that the RAL $XCH_4$ is even larger than the TCCON measurement in spring at high-latitude sites. The time series of the differences between the RAL IASI and NDACC measurements at 13 sites are also shown in Figure 16(right). The mean and stdv of the (discontinuity corrected) differences are 9.79 ppb and 15.26 ppb (uncorrected 9.55 ppb and 15.82 ppb), respectively. Similar to the TCCON comparison, for the low- and mid-latitude sites, it is found that there is little latitude dependence in the difference between RAL and NDACC measurements, and the differences are within $\pm$20 ppb. However, in high-latitude sites ($>60°$N) the RAL $XCH_4$ is 20-100 ppb systematically larger than the NDACC measurements at Thule, and the RAL $XCH_4$ is generally larger than the NDACC measurements in spring at Kiruna.

### 6.2.2 LMD bias and scatter

Figure 17 shows the time series of the differences between LMD IASI and the ground-based FTIR measurements between July 2007 and June 2015. Compared to RAL measurements, there are no available co-located LMD measurements at Eureka, Rikubestu and Edwards TCCON sites. The mean and stdv of the differences are -4.76 ppb and 16.32 ppb, respectively. It is noticed that the mean differences vary with latitude, with strong positive values in the tropical (Ascension, Darwin and Reunion) regions, but negative values in the mid-latitude region. The bias at high-latitudes tends to be neutral to lightly positive. Moreover, there is a strong seasonal variation in the difference. For example, at Lamont, the LMD IASI is about 20 ppb larger than the TCCON measurement in summer but it is about 60 ppb less than the TCCON measurement in winter. The mean and stdv of the differences between LMD and NDACC measurements are 2.83 ppb and 18.54 ppb, respectively. Similar to the TCCON measurements, the dependencies of the differences on latitude and time are also observed in comparison with the NDACC measurements.

A summary of the LMD and RAL comparison results with ground-based remote sensing FTIR measurements is given in Table 6.

### 6.2.3 RAL trend and seasonal cycle

The seasonal variations and long-term trends of $XCH_4$ observed by co-located anomaly-corrected RAL and TCCON measurements are shown in Figure 18. The $XCH_4$ trends derived from RAL are systematically lower than those derived from TCCON

**Table 6.** The mean station bias and mean station stdv of the difference between IASI satellite $CH_4$ and ground-based FTIR measurements (SAT-GB).

| Satellite | LMD | | RAL | |
|---|---|---|---|---|
| | mean | std | mean | std |
| TCCON | -4.76 ppb | 24.00 ppb | 5.05 ppb | 11.23 ppb |
| NDACC | 1.83 ppb | 26.46 ppb | 9.79 ppb | 15.26 ppb |

measurements at almost all sites. The mean $XCH_4$ annual growth rates, using only stations that cover a substantially long time window, are 4.84 ppb/yr derived from the RAL measurements, and 6.1 ppb/yr derived from TCCON measurements. Note that, due to the limited co-located RAL and TCCON measurements at Eureka, Rikubetsu and Edwards (Figure 16), the uncertainties of the trends at these sites are relatively large. In general, both the phase and amplitudes of the seasonal variations of $XCH_4$

observed by RAL and TCCON are close to each other. Again looking at the long-running core stations, the RAL-TCCON long-term trend bias difference, ranging between -2.72±0.62 ppb/year (Lamont) and -0.47±0.46 ppb/year (Lauder), shows little latitudinal dependence apart from the observation that Southern hemisphere RAL-TCCON trend differences are slightly smaller, compared to those observed at Northern hemisphere stations.

    The seasonal variations and long-term trends of $XCH_4$ observed by RAL and NDACC measurements are shown in Figure

19. Here the difference in long-term trend biases show more station to station variability compared to TCCON. No doubt in part because NDACC also retrieves (limited) profile information, but also because harmonization within TCCON is more rigorous. Only Wollongong and Reunion St-Denis underestimate the long-term trend with respect to NDACC with (slightly) more than 2 ppb. All other stations feature positive and negative biases within this 2 ppb range. One could discern a small latitudinal dependence with Southern hemisphere stations featuring on average slightly stronger negative RAL-NDACC trend biases, but

if present it is very small. Note that the inverse dependence is shown in the TCCON comparisons (~1 ppb stronger negative RAL-TCCON biases in the Northern hemisphere).

    The strongest difference is observed at the Maïdo station but here the uncertainty on the trends are very high since it only commenced measurements in 2013. The mean of the $XCH_4$ trends derived from the RAL data is 4.77 ppb/yr, which is only slightly less than that from NDACC measurements of 4.91 ppb/yr. The phases and amplitudes of the seasonal variations of

$XCH_4$ observed by RAL and NDACC are similar at most sites, which is consistent with the TCCON measurements.

    Of course, the long-term trend analysis is impacted by the May $16^{th}$ 2013 discontinuity and while we have applied a correction (+6.7 ppb shift) onto the data, our analysis in Section 5.2 also showed that, this corrected trend still ended up being ~1 ppb/year lower compared to an approach where the dataset was split in 2 independent sections. However regardless of the correction method used, RAL trends consistently (with little latitudinal dependence) underestimated the long-term trend

when compared to CAMS (see Figure 12), which is consistent with our TCCON comparisons (average RAL-TCCON trend difference of -1.85±0.85 ppb/yr). For NDACC we see both over and under-estimations of the trend depending on the station with on average a -0.14±1.24 ppb/yr RAL-NDACC long term trend difference.

### 6.2.4 LMD trend and seasonal cycle

The seasonal variations and long-term trends of mtCH$_4$ observed by LMD and TCCON measurements are shown in Figure 20. When regarding the core stations, the LMD-TCCON trend difference ranges between -4.48±1.41 ppb/yr (Lauder) and 0.91±1.35 ppb/yr (Darwin). In the Northern hemisphere alone, it ranges between -3.76±1.71 ppb/yr (Bialystok) and -1.09±1.65 ppb/yr (Orleans). Overall the bias differences are more outspoken compared to the RAL-TCCON trend differences. We also typically find the strongest negative LMD-TCCON trend biases outside the 40°S-40°N range (-4.48 ppb/year at Lauder, -3.76 ppb/year at Sodankyla), but the variability within and outside the 40°S-40°N range is considerable. For instance, the trend difference at Orleans (48°N, -1.09 ppb/yr) is smaller than that observed at Lamont (36.6°N, -2.36 ppb/yr).

If we only consider LMD's core latitude region (30°N-30°S) (Izana, Ascension, Darwin and Reunion), the mean and stdv of the mtCH$_4$ trends are 6.4±0.9 ppb/yr derived from TCCON measurements, and 6.0±1.8 ppb/yr derived from LMD measurements. Concerning the seasonal variation of the mtCH$_4$, the differences between the LMD and TCCON measurements are obvious at Bialystok, Karlsruhe, Garmisch (European sites), Park Falls, Lamont (American sites), Reunion and Lauder. For example, at Park Falls, the mtCH$_4$ observed by LMD is high in July and low in January, but the mtCH$_4$ observed by TCCON measurement is low in July and high in January. Moreover, the amplitude of the seasonal variation at Park Falls observed by LMD is about 80 ppb, which is 4 times larger than that observed by TCCON measurements of about 20 ppb. These differing seasonal patterns lead to the biases as observed in Figure 17 (LMD-TCCON and NDACC) and Figure 6 (top row, LMD-CAMS), with significant negative biases in autumn-winter and positive biases in summer over the United States and (less outspoken) Europe.

The seasonal variations and long-term trends of mtCH$_4$ observed by LMD and NDACC measurements are shown in Figure 21. Compared to our TCCON analysis, the station to station variability in the LMD-NDACC trend differences are substantially greater, ranging from -5.35±3.09 ppb/yr (Eureka, 80.1°N, Canada) to 5.98±3.11 ppb/yr (St-Denis, 20.9°S, Reunion (France)). St.Petersburg aside, one would see a very clear latitudinal dependence with LMD underestimating the long term trend at high latitudes and overestimating them near the (sub) tropics. Note that in our analysis of the long term trend using CAMS data (see Section 5.2 and Figure 12), we saw a stable but slightly underestimated long-term trend in the (sub)tropics (roughly between 40°S and 40°N), with a rapidly increasing underestimation at higher latitudes.

The seasonal variations of mtCH$_4$ observed by LMD and NDACC are similar at Kiruna, Izana, Mauna Loa and Wollongong. However, the seasonal variations of mtCH$_4$ observed by LMD and NDACC are different at Garmisch, Zugspitze, Jungfraujoch (European sites), and Reunion. Both TCCON and NDACC measurements suggest that there is large uncertainty in the seasonal variation of mtCH$_4$ observed by LMD in Europe and Reunion.

Table 7 below shows the averaged (over all core stations) long term trends of both LMD and (bias-corrected) RAL and their corresponding co-located NDACC and TCCON mtCH$_4$ timeseries. Overall, on average, both LMD and RAL underestimate the long term trend. Also immediately apparent is the far greater standard deviation on the trend for LMD, compared to RAL, indicating stronger station-to-station variability.

**Table 7.** Mean and standard deviation of averaged (over long-running stations only) long term trends differences (in ppb/yr) between IASI satellite and ground-based FTIR $CH_4$ measurements (SAT-GB). RAL measurements have been corrected for the discontinuity issue by using a +6.7 ppb bias correction

| Satellite | LMD | | GB | | RAL | | GB | |
|---|---|---|---|---|---|---|---|---|
| | mean | std | mean | std | mean | std | mean | std |
| TCCON | 4.26 | 1.27 | 6.31 | 0.81 | 4.84 | 0.47 | 6.10 | 0.74 |
| NDACC | 3.53 | 2.81 | 4.64 | 1.76 | 4.77 | 0.63 | 4.91 | 1.07 |

## 6.3 Short summary

Of the 3 *in situ* measurement data used, AirCore, which measures profiles well into the stratosphere, can be considered the most representative. Unfortunately while the LMD-AirCore bias is lower than that of RAL, its very limited dataset does not warrant a definitive conclusion as to the observed bias differences in the direct comparisons. Certainly in view of the rather significant impact of, and the uncertainty associated with, the May 16[th] correction (RAL-AirCore bias from -10.2 to -5.9 ppb after correction).

Looking closer at the HIPPO measurements we see that the biases with respect to HIPPO are far more outspoken in the LMD data, as compared to RAL. However we do not see ever more negative LMD-HIPPO biases when moving towards the polar regions (as hinted at in Figure 12), in stead we the strongest negative biases occur around 40-50°N. Note that HIPPO measurements are concentrated around the Pacific Ocean area (Figure 1) only and thus do not yield a global picture of the quality. However they do cover a wide range of latitudes and cover (although not in the same year) all seasons. Looking at the region in more detail, we observe that the more outspoken bias outliers in Figure 14 spatially (looking at the Pacific Ocean latitude band) and temporally (looking at the season) correspond closely with the areas that exhibit stronger biases in the CAMS-LMD comparisons (Figure 6 top row). For instance, between 40°N and 50°N we see many strong negative LMD-HIPPO biases which correspond with the often observed stark negative LMD-CAMS bias over this area, particularly in October-January. On the other hand, around 30°S LMD-CAMS features often strong positive biases, again corresponding with the values found in the LMD-HIPPO comparisons.

RAL-CAMS biases (Figure 6 second row) over the Pacific Ocean, on the other hand are not as strong which is again reflected in the HIPPO comparisons. The IAGOS measurements, due to their irregular and often limited spatial and temporal distribution combined with inherent scatter didn't allow us to determine whether the observed RAL-LMD differences (Figure 6 bottom row) can be attributed to either algorithm. It is also very important to note that both HIPPO and IAGOS need extending the aircraft *in situ* profiles with CAMS model data. Particularly potential model errors in the exact location of the sharp $CH_4$ concentration decrease as one ascends through the Upper Troposphere-Lower Stratosphere (UTLS) can have a significant impact which is hard to quantify with no exact information on the true state of the atmosphere. Therefore HIPPO's apparent corroboration of the Satellite-CAMS comparison results should be interpreted with extreme caution. As a test we artificially lowered or heightened the point at which the UTLS $CH_4$ CAMS transition kicks in by 100 hPa. This nullified most but not all of the most outspoken

biases between LMD and HIPPO. Of course this would imply that the direction of the hypothetical correction needs to change in sync with the latitudinal pattern in Figure 14 as it features positive and negative biases alike.

The ground-based FTIR (TCCON and NDACC) measurements are used to compare with two IASI $CH_4$ products of RAL and LMD (Tables 6 and 7). The TCCON and NDACC measurements show that the systematic uncertainties of RAL and LMD data are both within $\pm 10$ ppb. However, the stdv of the differences between LMD and FTIR is about 25 ppb, which is larger than that between RAL and FTIR of about 11-16 ppb. While the limited number of stations, the uncertainty on the individual station biases, and the considerable station-to-station variability make it impossible to definitively prove a strong latitudinal dependence of LMD's long-term trend, the comparisons with TCCON and NDACC certainly do not run contrary to our LMD-CAMS analysis in Section 5.2. The fact that the RAL-TCCON and RAL-NDACC trends do not feature such a latitudinal dependence further corroborates our analysis. We also observe significant differences at several sites with respect to the seasonal cycle in both TCCON and NDACC.

Here we need to add that there are very little stations within the 30°N-30°S latitude band for which the LMD algorithm was initially targeted. Also note that TCCON uses a profile scaling retrieval approach in which the shape of the *a priori* profile cannot be altered resulting in potential smoothing errors.

# 7    Discussions

## 7.1    Two partial columns derived from RAL

In Section 6.1, it is found that the RAL $XCH_4$ is about 16.5 ppb underestimated between 15°N and 15°S as compared to HIPPO measurements, and the underestimation is mainly coming from the lower partial column (0-6 km). The DOFs of RAL indicate that, apart from the higher latitudes (>60° North and South), 2 independent partial columns can be obtained from the retrieved profiles. Unfortunately, the reference dataset remains fairly limited with regards to accurately assessing partial column information. Not enough vertical profile information is available in the ground-based FTIR measurements. As for the *in situ* observations, they have limited temporal-spatial coverage.

However, our analysis of the RAL upper-lower partial column differences compared to CAMS's partial column differences (see Section 4.2) equally showed a pronounced RAL lower layer $qCH_4$ underestimation in the Pacific Ocean between 15°N and 15°S, that was 12.5 ppb less than the CAMS model. This is consistent with the comparison between RAL and HIPPO in this lower layer.

Apart from the uncertainties within the CAMS model, there might be many reasons for the observed partial column differences from the RAL retrieval such as the uncertainty of the spectroscopy, meteorological parameters etc., which could potentially affect partial columns differently. In addition, optimal estimation retrievals rely on a fine balance between placing too much constraint on the retrieval, resulting in too little retrieval information being added to the *a priori* and thus lower degrees of freedom on the one hand and placing not enough constraints on the retrieval, which risks producing unrealistic retrieval results. The latter often presents itself most clearly in unrealistic vertical retrieval profiles. Other observations that indicate a large sensitivity on the measured radiances are the 7 to 8 ppb bias between IFOV 1 and IFOV 3 of IASI, and the stark contrast

between the upper and lower partial column bias between adjacent land and sea measurements. If true, additional constraints need to be added to the retrieval, thereby adding stability at the cost of degrees of freedom, potentially losing the capacity to resolve two independent layers. Another factor that might be at play is the limited vertical resolution of the retrieved profile and associated averaging kernel. Both partial columns effectively correspond with 1.5 layers in the profile, which leaves little room for accurately capturing any potential variability in the sensitivity within each layer, nor does it ensure true independence between the partial columns. Further investigation is needed to understand the performance of the RAL two partial columns better, when more *in situ* data become available.

## 7.2 LMD seasonal cycle discussion

By comparing LMD with RAL, CAMS, HIPPO and ground-based FTIR measurements, it is found that the seasonal variation of mtCH$_4$ observed by LMD is different from others, especially in certain latitude regions (see Figure 6). However CAMS model data can hardly be regarded as the true state of the atmosphere, only an approximation thereof. And while HIPPO measurements are highly accurate, they need to be expanded by model profiles to cover LMD's entire vertical sensitivity range. Changing the CAMS UTLS transition region resulted in significant changes in the observed biases with HIPPO. However in most cases, the biases increased instead of decreased and in the rare cases the comparison improved, upward shifts of up to 4 km of the transition region were required. Ground-based remote sensing TCCON FTIR measurements does not need model profile extensions but it uses a profile scaling retrieval approach and since the shape of the profile is of great influence when applying LMD's sensitivity profile, one could certainly cast doubt on these observed biases as well. One can only point out that they confirm the HIPPO and CAMS observations even though they use a different approach to construct their *a priori* profile shape. NDACC FTIR retrievals on the other hand use an optimal estimation approach which allows for profile shape optimization and here again we observe seasonal bias variability at, for instance, the Jungfraujoch data. Unfortunately its vertical profile resolution is very limited (DOFs is about 2.5) and one could claim that this is insufficient for an accurate application of the LMD sensitivity profile. Therefore in this section, we use the AirCore profiles at Boulder (39.7°N, 104.8°W) as the reference data to compare with the CAMS model and the LMD measurements. The AirCore profiles at Boulder are selected because the seasonal variations of CH$_4$ observed by LMD and RAL are very different at North American Template region (Figure 10). There are 5 AirCore measurements available at Boulder between October 2017 and September 2019. Figure 22 shows the AirCore vertical profiles, and the mtCH$_4$ derived from these AirCore profiles with the smoothing correcting using the LMD weighting function. The AirCore profile has a good vertical coverage, providing measurements between the surface and the stratosphere (about 25 km) so no (potentially flawed) model data is required to extend its profile over the troposphere-stratosphere boundary where a sharp decrease in CH$_4$ occurs. The seasonal variation of mtCH$_4$ derived from the LMD measurements within ±5° latitude and ±5° longitude around Boulder between June 2007 and June 2015 show that mtCH$_4$ is high in summer and low in winter. However, the AirCore measurements show that the mtCH$_4$ is low in summer and high in autumn and winter, which is consistent with the TCCON measurements at Lamont (Figure 20). The CAMS model at Boulder shows a seasonal cycle phase that is in line with the AirCore measurements although its amplitude looks underestimated. Since we see a clear latitudinal dependence of both the long term trend as well as the seasonal cycle offset we have likewise obtained the long term trend and seasonal

cycles of the SAT-CAMS biases, grouped per 10° latitude band. The results thereof are shown in Figure 12. It shows that the amplitude of the seasonal cycle in the RAL-CAMS residuals is consistently lower than for LMD. Note that, as shown above, the CAMS seasonal cycle may not be accurate itself. However what is more of interest is that we see a strong increase in the LMD residual seasonal amplitudes at higher (>50°) latitudes. Likewise for the long term trend, RAL (both versions) shows little variability in the long term trend of the RAL-CAMS residuals, whereas LMD shows a very strong decrease in the residual trend at latitudes exceeding (40°) in both hemispheres.

To conclude, while we accept that the used dataset is very limited, combined with all the other observations, AirCore measurements strongly suggest that the seasonal variation of mtCH$_4$ observed by LMD retrievals have a significant overestimation of the seasonal amplitude together with a misrepresentation of the phase and an underestimation of the long term trend above several higher-latitude regions. This observed decrease in fidelity at higher latitudes calls for an investigation of the robustness of LMD's Neural Network training database for these scenes. This is acknowledged by the algorithm development team as it currently advises users to be cautious when handling data from latitudes beyond 60° North and South. Our analysis however shows that even at lower latitudes timeseries start becoming less robust as compared to the 35°N-35°S latitude band.

## 8   Conclusions

The goal of this study was to perform an extensive validation of two IASI global CH$_4$ products (RAL and LMD) between July 2007 and June 2015, using a wide array of reference measurements.

The IASI products are compared to *in situ* and ground-based FTIR data. Average differences with respect to *in situ* measurements for LMD range between -0.3 and 10.9 ppb, while for RAL (discontinuity corrected using a +6.7 bias shift) they range between -4.6 and -1.6 ppb. For the *in situ* comparisons, the differences from RAL are consistently more negative than those from LMD, but in varying degrees. The stdv of the observed differences are consistently smaller for RAL. For AirCore these differences in stdv are small (15.2 ppb for LMD vs. 14.1 ppb for RAL). For IAGOS and HIPPO, these differences are more substantial. Moreover, it is found that there is about 16.5 ppb underestimation in XCH$_4$ for RAL measurement in the tropics, which is mainly coming from the lower layer between 0 and 6 km. Using the ground-based FTIR sites as the reference data, the mean stdv of the differences at the ground-based stations show significantly lower values for RAL (11-16 ppb) than those for LMD (about 25 ppb). Looking at the latitudinal and seasonal variability at TCCON and NDACC sites, we observe that RAL shows little latitudinal dependence, while LMD data is on average larger than TCCON measurements in the tropical region and smaller than TCCON measurements at mid- and high-latitude sites.

An analysis of the long-term trend and seasonal cycles of the LMD and RAL products was carried out. We observed significant differences between the two algorithms. For RAL, we initially observed a significant underestimation of the long-term trend. This is due to an anomaly occurring on 16 May 2013 (due to a change in the IASI level 1 product) which caused a significant bias shift. The L1 discontinuity in May 2013 is expected to be resolved in a future updated version of the RAL XCH$_4$ product, by using reprocessed L1 data. For LMD we observed significant deviations (with respect to RAL and the reference data), of the seasonal cycle (both in the magnitude of the amplitude as well as its phase) over several higher (>35°)-latitude

regions. We also found an underestimation of the long-term trend at higher latitudes. All in all this results in large seasonal biases at these sites during the later years of the timeseries.

Users should also be aware that, while the RAL partial columns manage to capture global features, they also still exhibit significant systematic errors. This observation, combined with the sensitivity of the retrieval with respect to the IASI L1 data

and detector IFOV, also poses the question whether the RAL Optimal Estimation retrieval requires more constraint. On the other hand, imposing a stronger prior constraint would result in more accurate retrieved values only if the "true" $CH_4$ distribution was adopted as the prior. Improvements to the scheme used to produce the data, which have been evaluated here, are ongoing and will be implemented for the next full reprocessing.

*Data availability.* The RAL data is publicly available at https://catalogue.ceda.ac.uk/uuid/f717a8ea622f495397f4e76f777349d1 (last access:

12 January 2023). The LMD data is publicly available at https://iasi.aeris-data.fr/ (last access: 12 January 2023). The HIPPO data is publicly available at https://www.eol.ucar.edu/field_projects/hippo (last access: 12 January 2023). The IAGOS data is publicly available at http://www.caribic-atmospheric.com/ (last access: 12 January 2023). The AirCore data is publicly available at ftp://aftp.cmdl.noaa.gov/pub/colm/AirCore/ (last access: 12 January 2023). The TCCON data is publicly available at https://tccondata.org/ (last access: 12 January 2023). The NDACC data is publicly available at https://ndacc.larc.nasa.gov/ (last access: 12 January 2023).

*Author contributions.* BD acted as the project PI. The validation work was carried out by BD, MZ, PP and YK with support from BL, CCP, CS and MDM. BK and RS provided RAL data and interacted with the validation team. MZ and BD prepared the paper. All authors contributed to the discussion and revision of the paper.

*Competing interests.* The authors declare that they have no conflict of interest.

*Acknowledgements.* This work was supported by the EUMETSAT Independent validation of $CH_4$ products (ITT 18/205) project. MZ would

also like to thank the National Natural Science Foundation of China (42205140) for additional support. We would also like to thank all the teams that carried out the reference data measurements (TCCON, NDACC, IAGOS, HIPPO and AirCore) and their respective funding agencies. CAMS model fields were generated using Copernicus Atmosphere Monitoring Service Information [2021]. Note that neither the European Commission nor ECMWF is responsible for any use that may be made of the Copernicus information or data it contains.

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

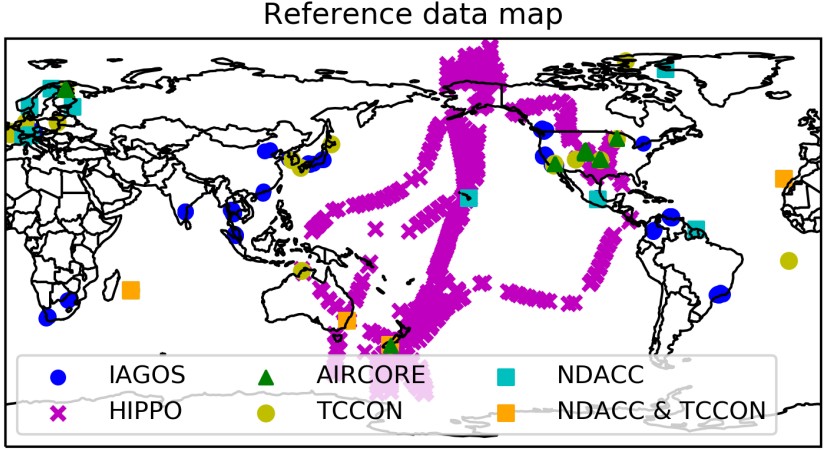

**Figure 1.** The map of the reference data used in this study, including *in situ* profiles and ground-based FTIR measurements.

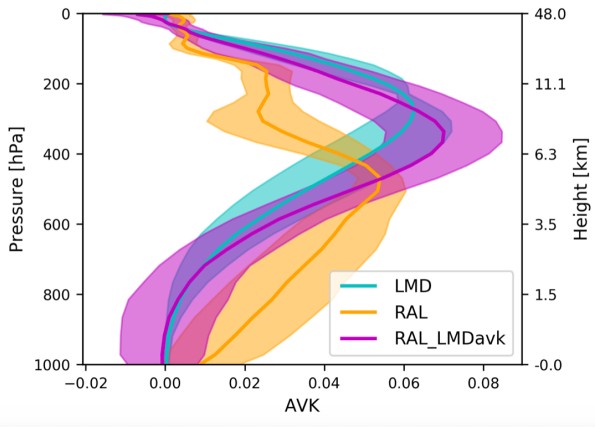

**Figure 2.** The vertical sensitivities of LMD, RAL and RAL but smoothed with the column averaging kernel of LMD (RAL_LMDavk). The solid line is the global annual mean in 2014, and the shadow is the standard deviation of all the averaging kernels in 2014.

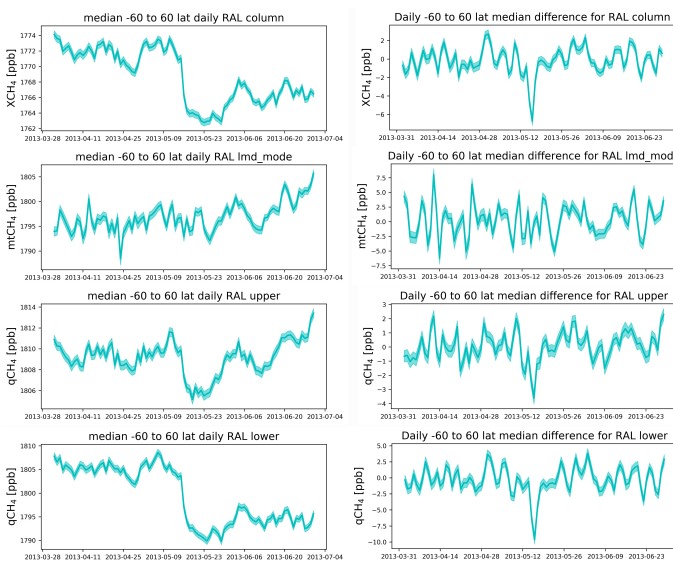

**Figure 3.** Left: Evolution of the median $CH_4$ concentration around the $16^{th}$ of May 2013, for the -60° to 60° latitude band for RAL $XCH_4$ (top), RAL_LMDavk $mtCH_4$ (second row), RAL (6-12 km) $qCH_4$ (third row) and RAL (0-6 km) $qCH_4$ (last row). Right: The difference between the median -60° to 60° concentrations before and after a given date for RAL $XCH_4$ (top), RAL_LMDavk $mtCH_4$ (second row), RAL (6-12 km) $qCH_4$ (third row) and RAL (0-6 km) $qCH_4$ (last row)

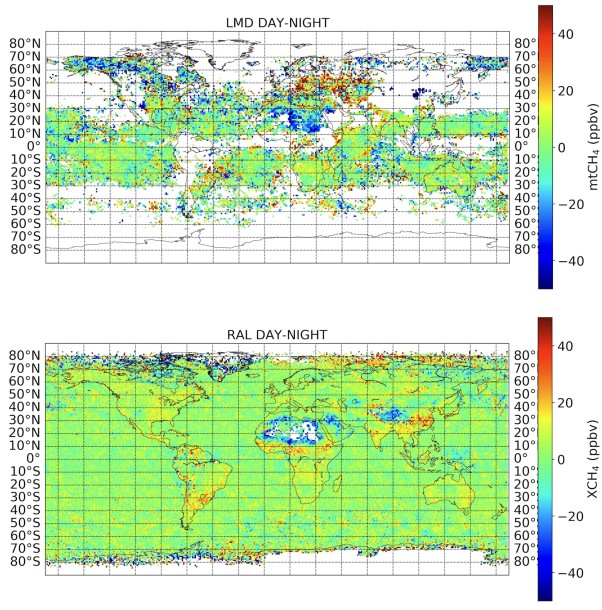

**Figure 4.** Monthly averaged LMD Day-Night $mtCH_4$ (top) and column averaged RAL (bottom) Day-Night $XCH_4$ differences for April 2012

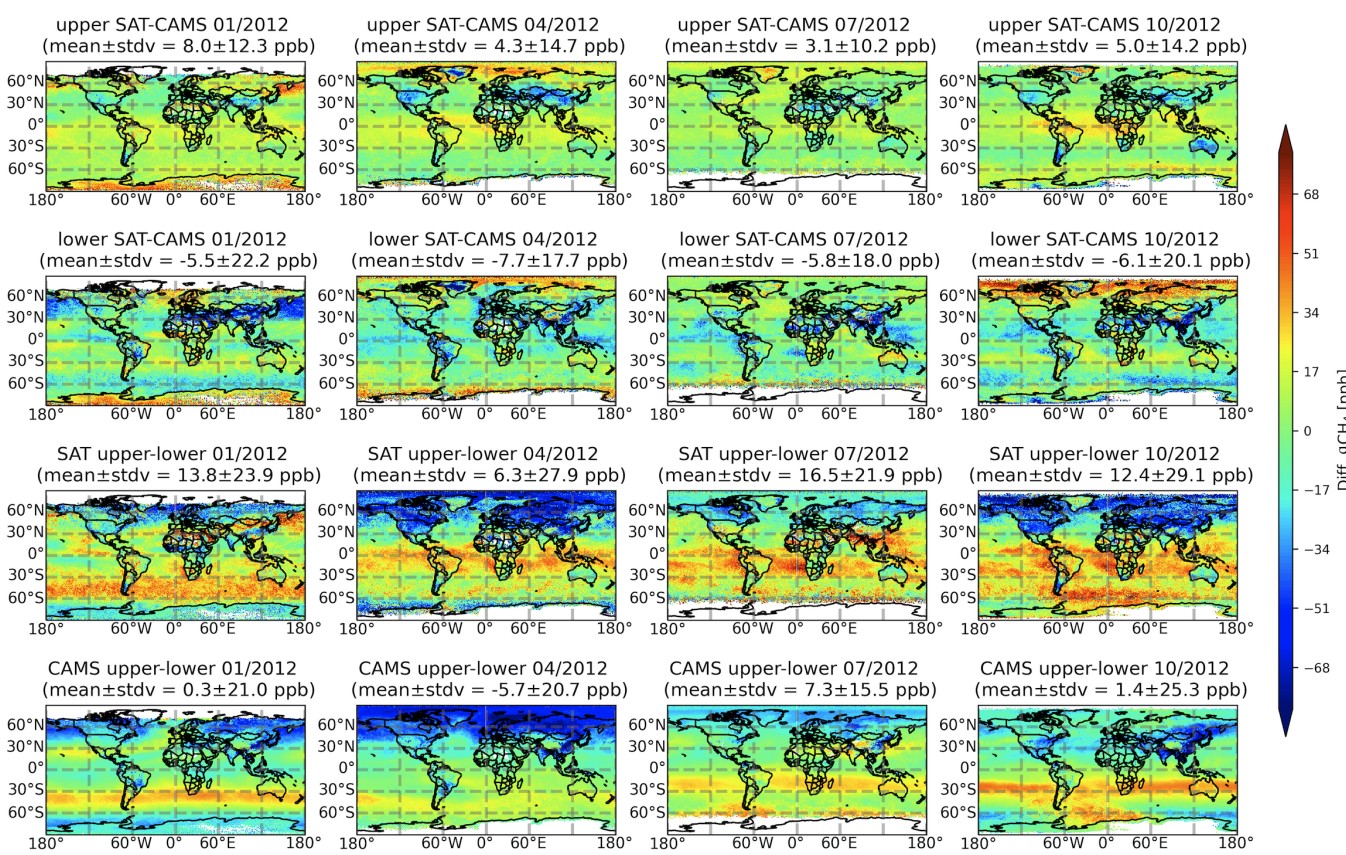

**Figure 5.** The difference of the qCH$_4$ in the upper layer (first row) and the lower layer (second row) between the RAL (SAT) and CAMS. Besides, the differences between the upper qCH$_4$ and lower qCH$_4$ derived from RAL (third row) and CAMS (last row) in January, April, July and October 2012.

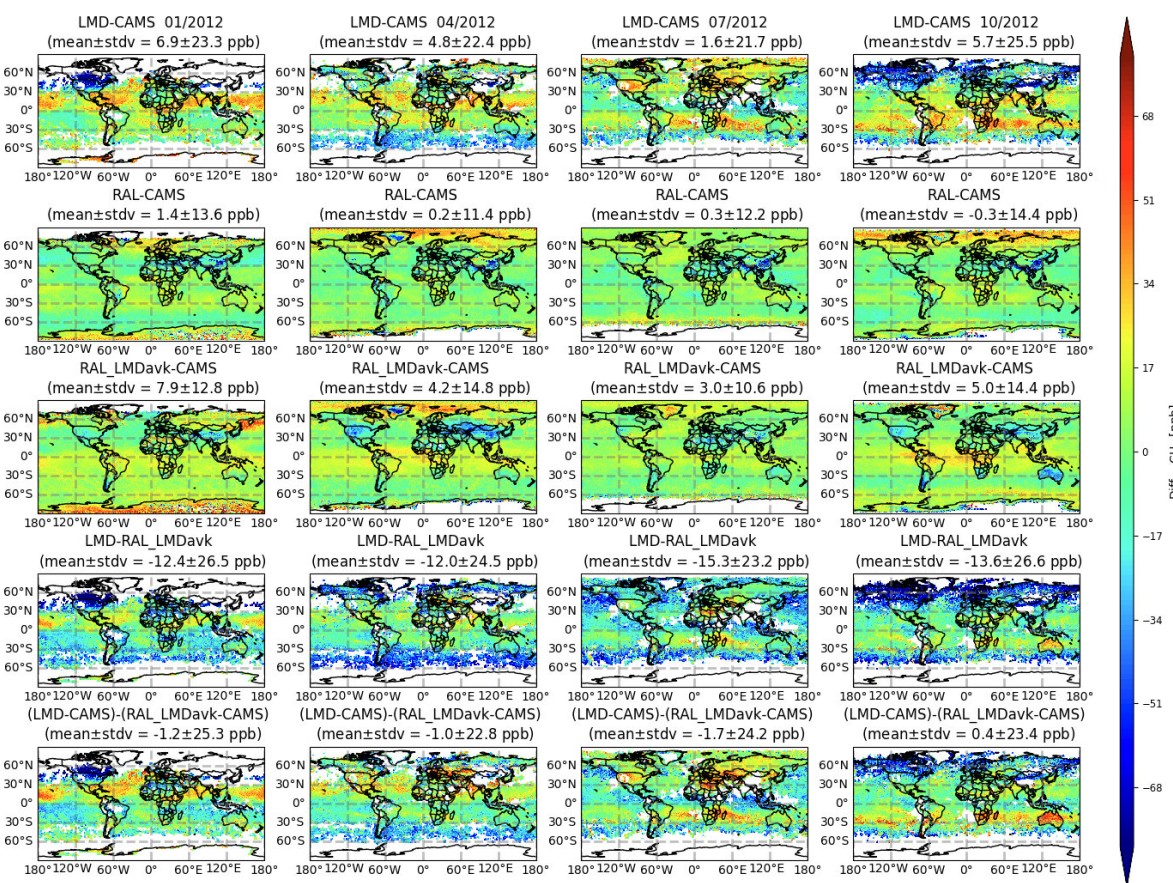

**Figure 6.** Global monthly mean maps for January, April, July and October 2012 (from left to right column). The top row shows the LMD-CAMS daytime mtCH$_4$ difference, the second and third row show the same but for RAL and RAL_LMDavk X(mt)CH$_4$ respectively. The last three rows show the intercomparison between the satellite products. Row 4 shows the LMD-RAL difference, row 5 LMD-RAL_LMDavk, while the last two rows compare the respective differences of LMD and RAL_LMDavk to their respective CAMS fields.

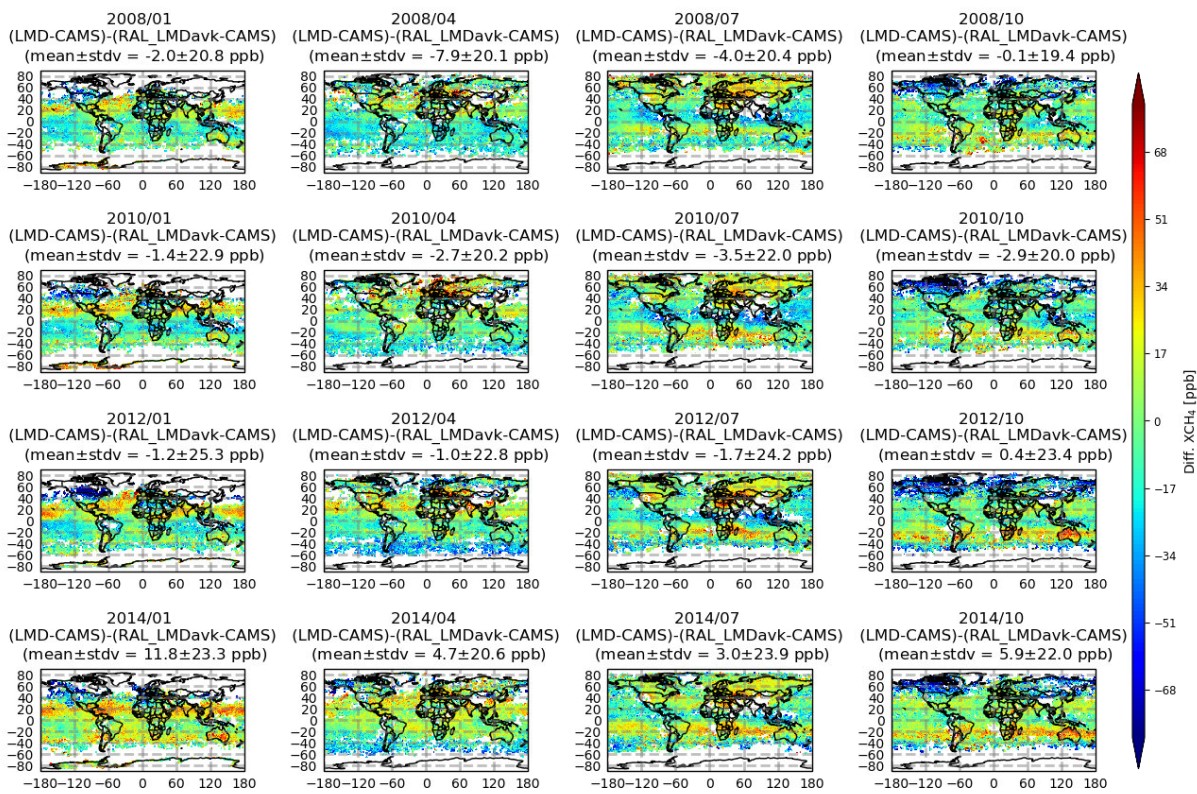

**Figure 7.** Global monthly mean maps for January, April, July and October (from left to right column) of the respective differences of LMD and RAL_LMDavk to their respective CAMS fields, for 2008,2010,2012 and 2014 (rows).

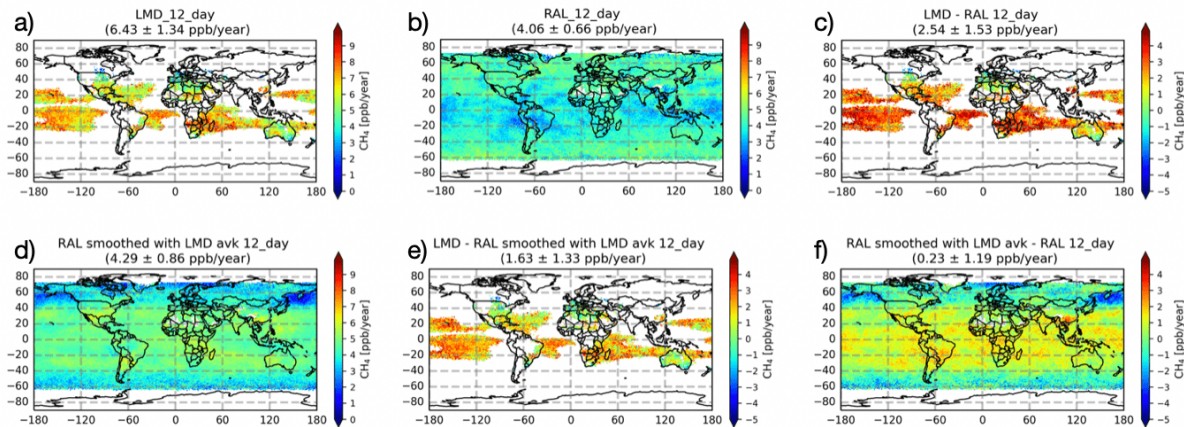

**Figure 8.** The mt(X)CH$_4$ annual growth derived from LMD (a), RAL (b), RAL_LMDavk (d) daytime measurements, together with their difference between LMD and RAL (c), between LMD and RAL_LMDavk (e), and between RAL and RAL_LMDavk (f). The CH$_4$ annual growth is only calculated for grid-boxes with less than 12 months of missing data between July 2007 and June 2015. No RAL discontinuity correction was applied.

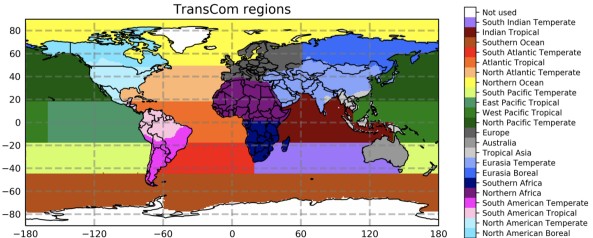

**Figure 9.** The TransCom map, including 11 land regions and 11 ocean regions.

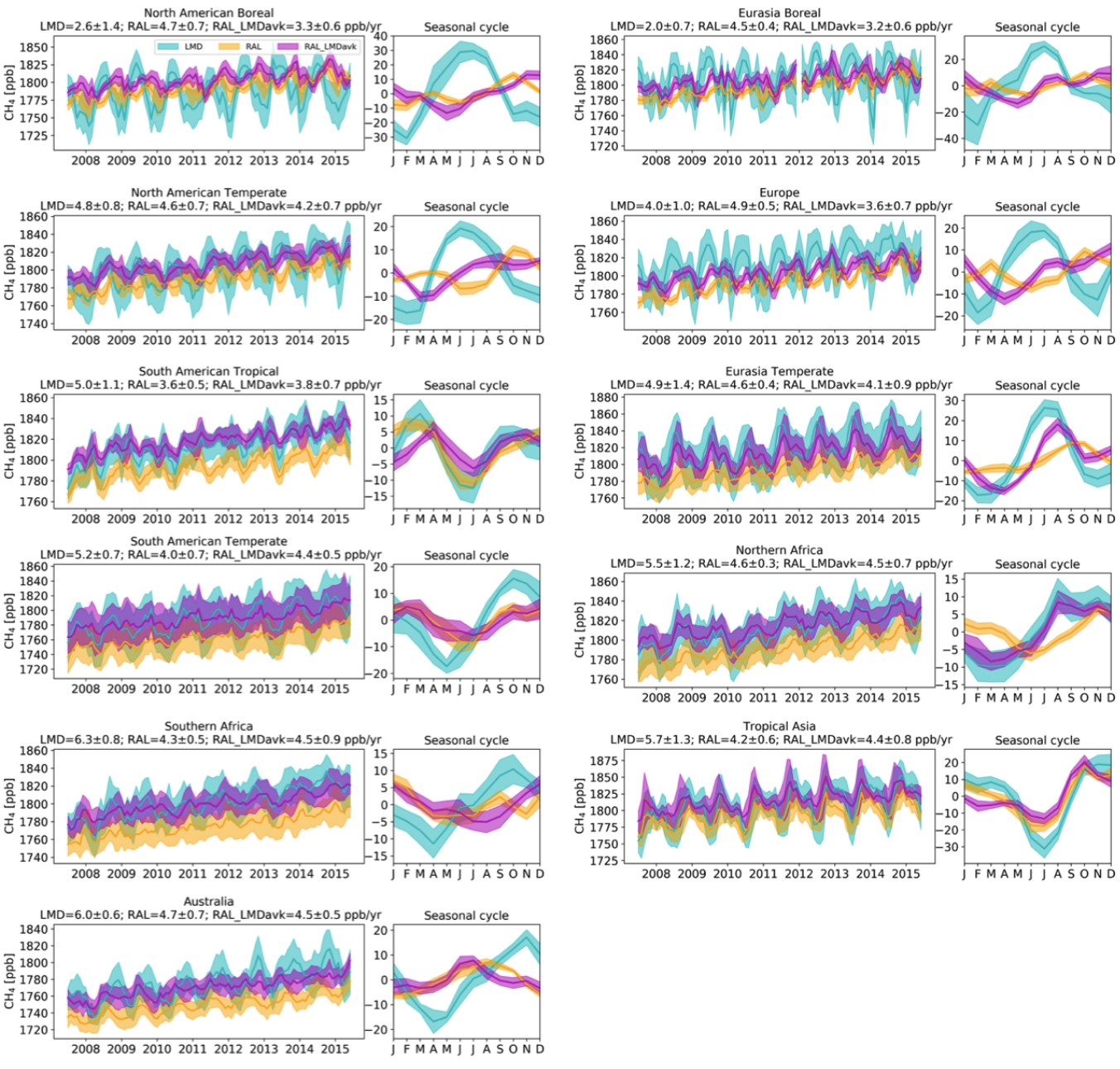

**Figure 10.** The time series of the LMD, RAL (+6.7 ppb discontinuity correction applied), and RAL_LMDavk CH$_4$ monthly means (solid lines) and standard deviations (shadow), together with the seasonal variations of CH$_4$ at 11 land TransCom regions.

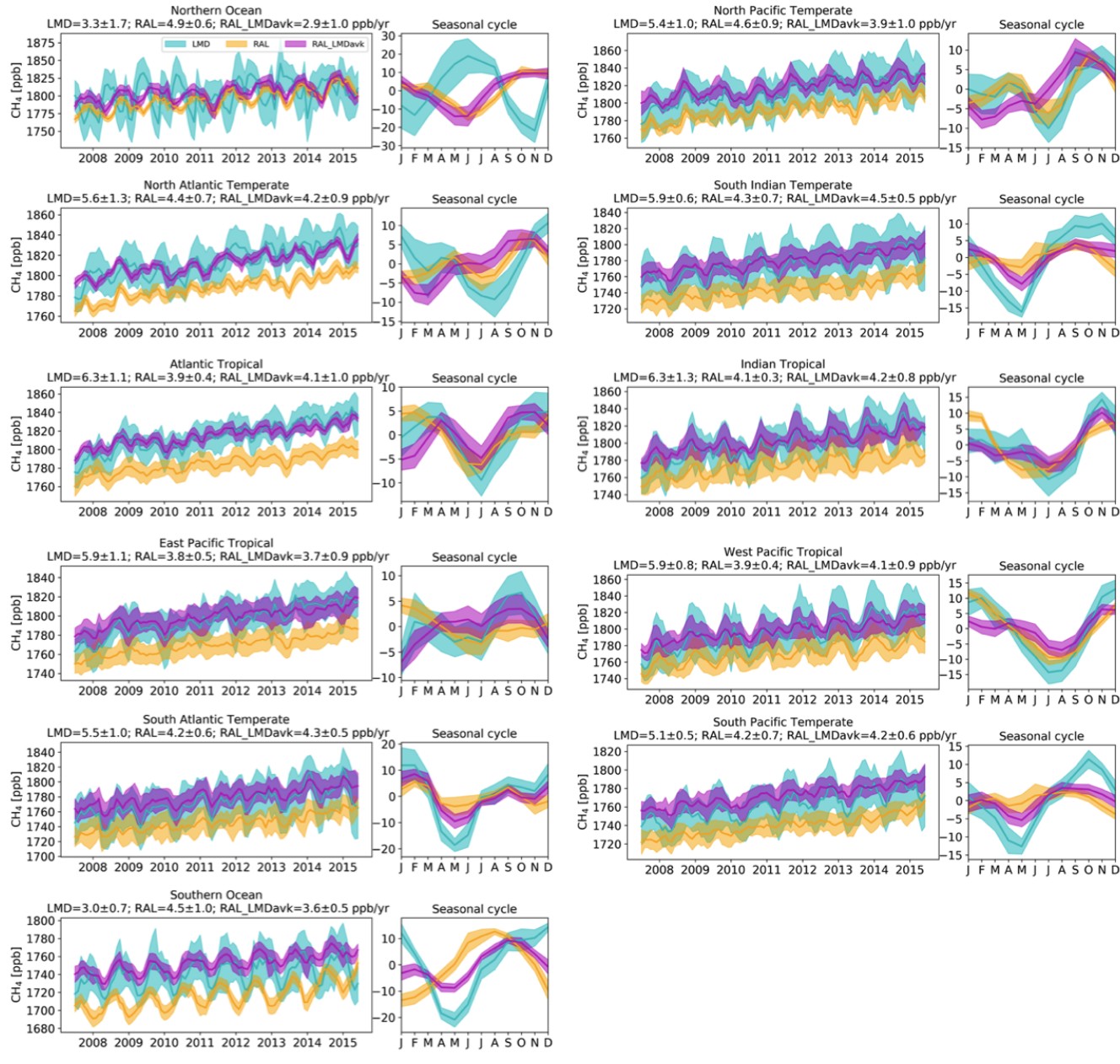

**Figure 11.** Same as Figure 10, but at 11 ocean TransCom regions.

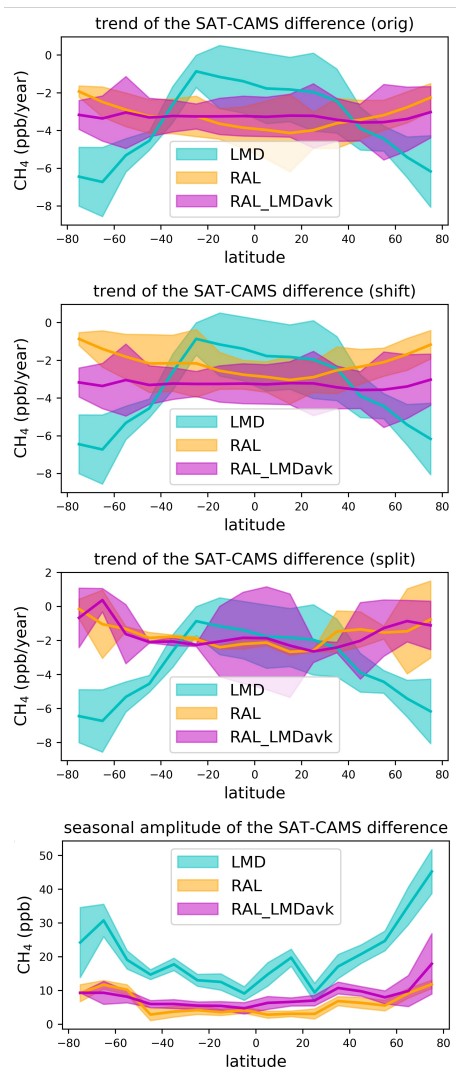

**Figure 12.** Top: long term trend values (ppb/year) for the satellite-CAMS residuals for $10°$ wide latitude bands for discontinuity uncorrected measurements. Second: Same as above but now with a +6.7 ppb RAL discontinuity correction, Third: Same as the top figure but now with RAL and RAL_LMDavk split into a pre- and post- May $16^{th}$ 2013 timeseries. The overall trend values corresponds with the time-weighted average of the trends from these 2 timeseries. Bottom: Seasonal cycle amplitude of the satellite-CAMS residuals for $10°$ wide latitude bands

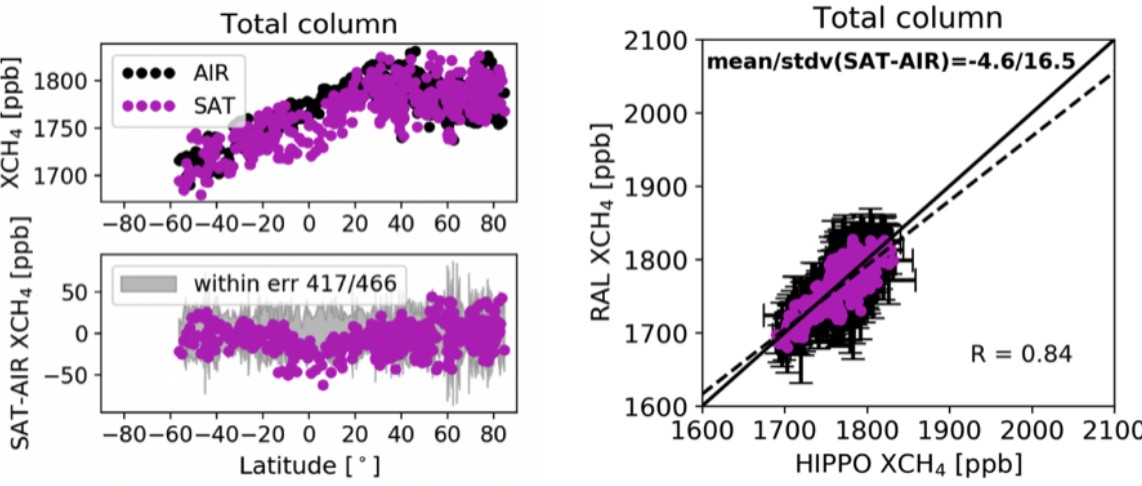

**Figure 13.** RAL (no data after May 2013) and HIPPO XCH$_4$ together with their differences at different latitudes (left), and the scatter plot between the RAL and HIPPO XCH$_4$ (right). R is the correlation efficient. The dotted and solid lines correspond with a linear fit through the data and a y=x line, respectively.

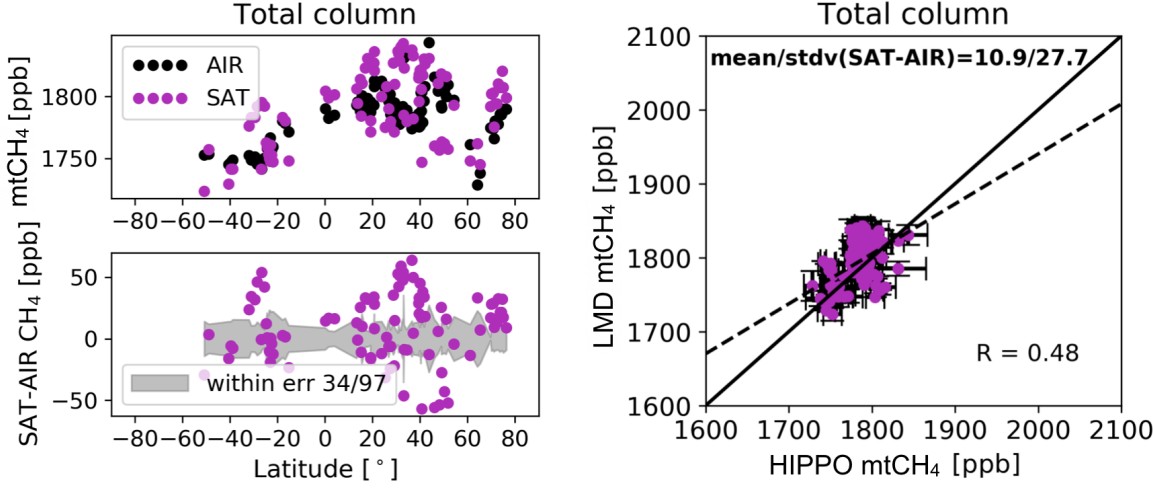

**Figure 14.** Same as Figure 13, but for LMD and HIPPO measurements.

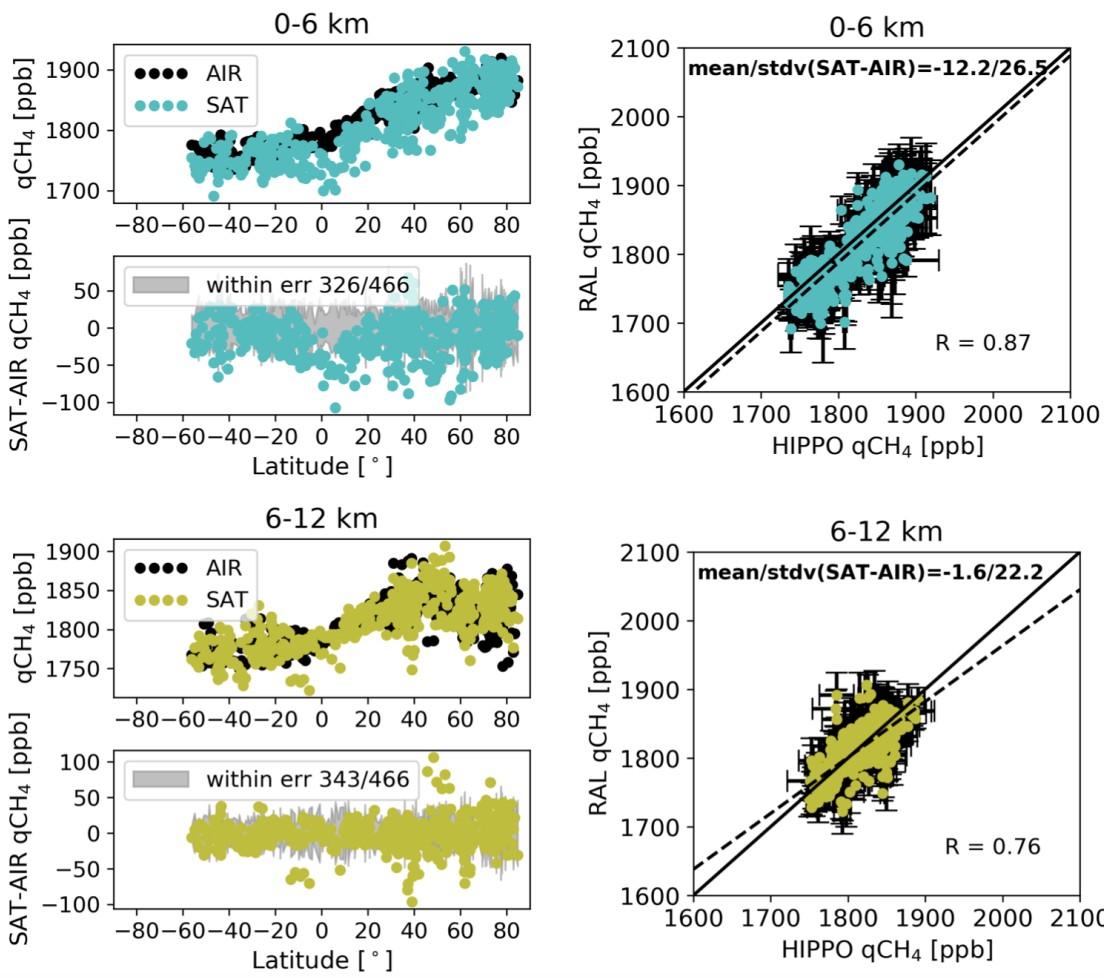

**Figure 15.** RAL and HIPPO qCH$_4$ together with their differences at different latitude (left), and the scatter plot between the satellite and HIPPO qCH$_4$ in the vertical ranges between 0 and 6 km (upper panels, no data after May 2013), and between 6 and 12 km (lower panels, no data after May 2013) (right). R is the correlation efficient. The dotted and solid lines correspond with a linear fit through the data and a y=x line, respectively.

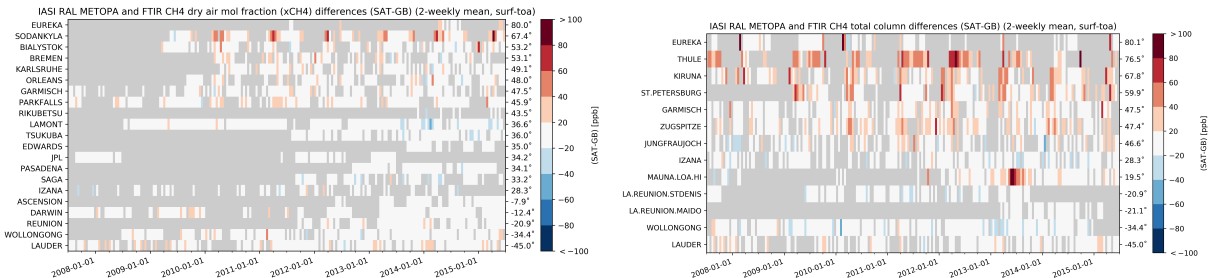

**Figure 16.** Mosaic plot of 2-weekly absolute mean differences (SAT-GB) at ground-based FTIR sites for the column averaged dry air mole fractions XCH$_4$ between RAL (no discontinuity correction applied) and ground-based FTIR measurements (left: TCCON; right: NDACC). The FTIR sites are sorted by their latitudes from north to south.

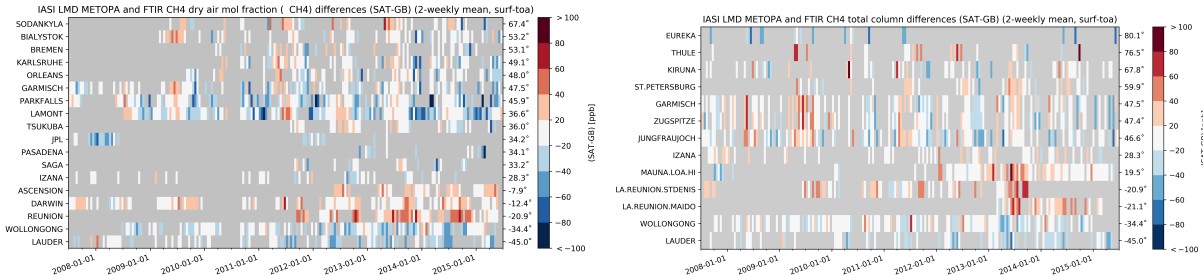

**Figure 17.** Same as Figure 16, but for mtCH$_4$ from LMD and ground-based FTIR measurements.

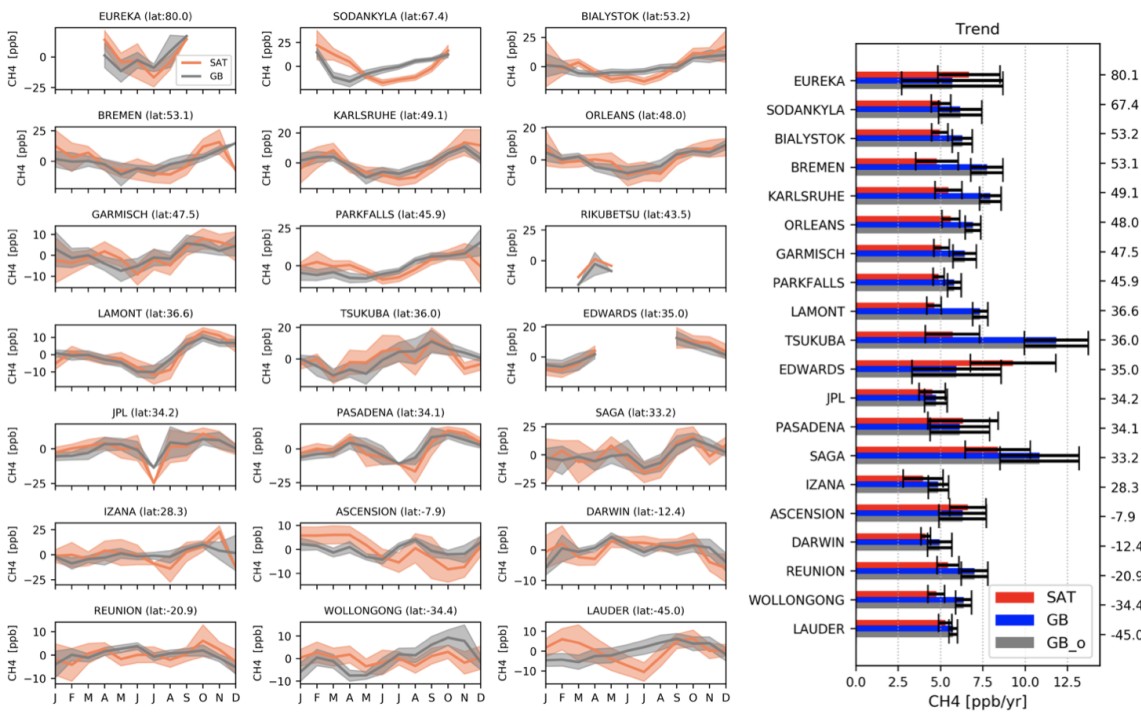

**Figure 18.** Left panels: the seasonal variations of XCH$_4$ observed by RAL (SAT) and smoothed TCCON (GB) measurements at each site. A +6.7 ppb discontinuity correction has been applied to the RAL data. Right panel: the XCH$_4$ annual growths derived from (discontinuity corrected) RAL observations (SAT), TCCON measurements after smoothing (GB) and original TCCON measurements (GB_o). The latitude of the TCCON site is also remarked in the title or the y-axis. Not enough Rikubetsu data pairs (only 8) were available to calculate a long-term trend.

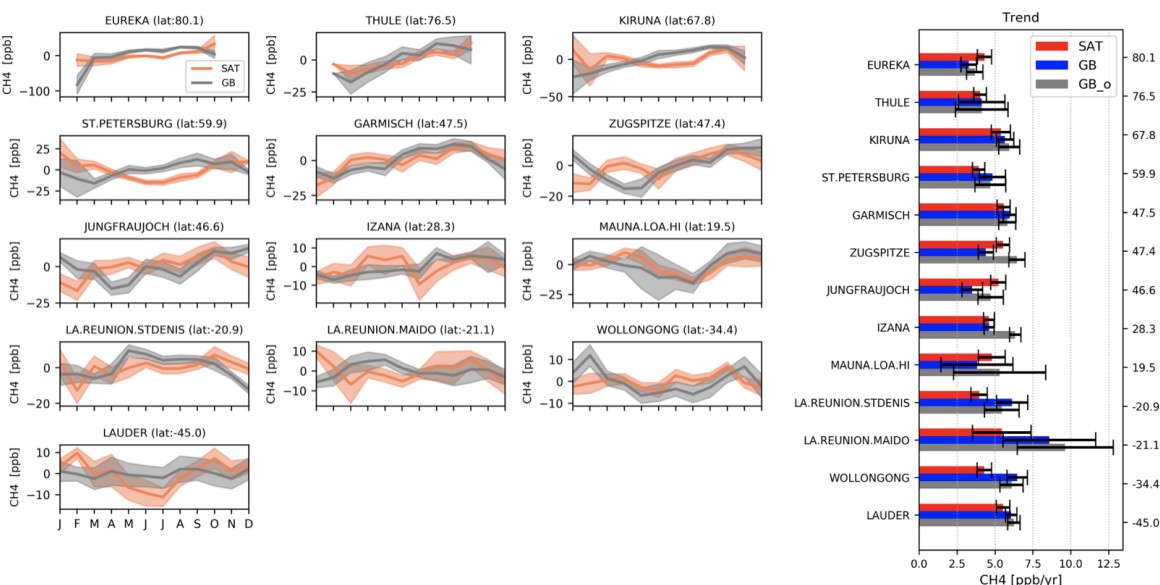

**Figure 19.** Same as Figure 18, but for RAL and NDACC measurements.

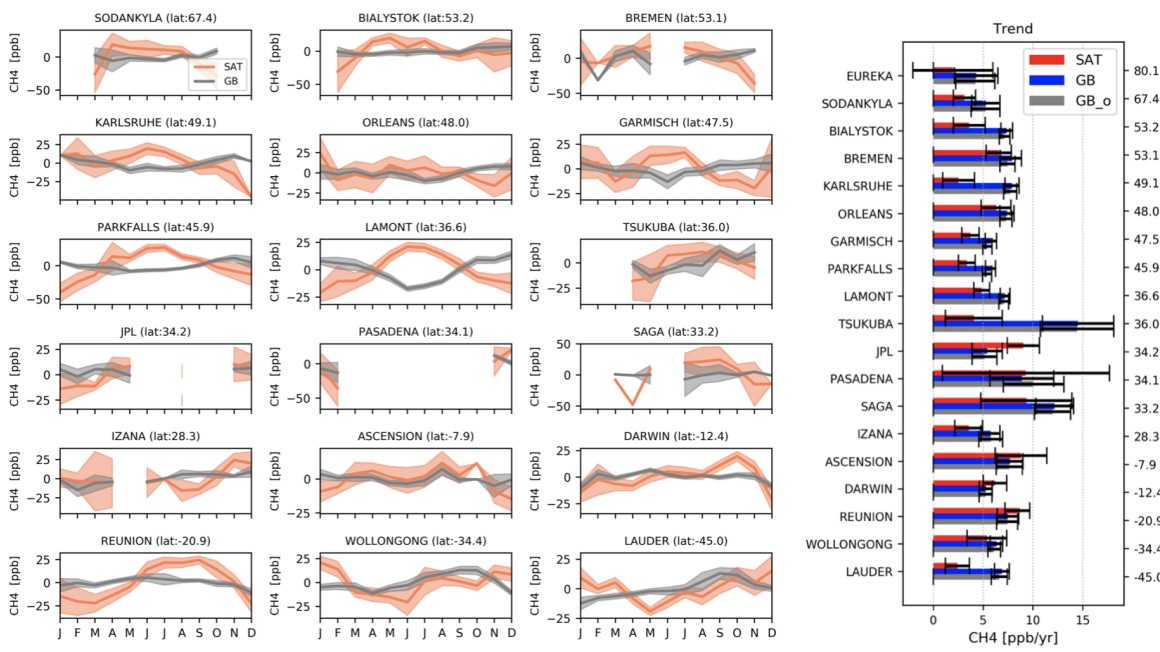

**Figure 20.** Same as Figure 18, but for LMD and TCCON measurements.

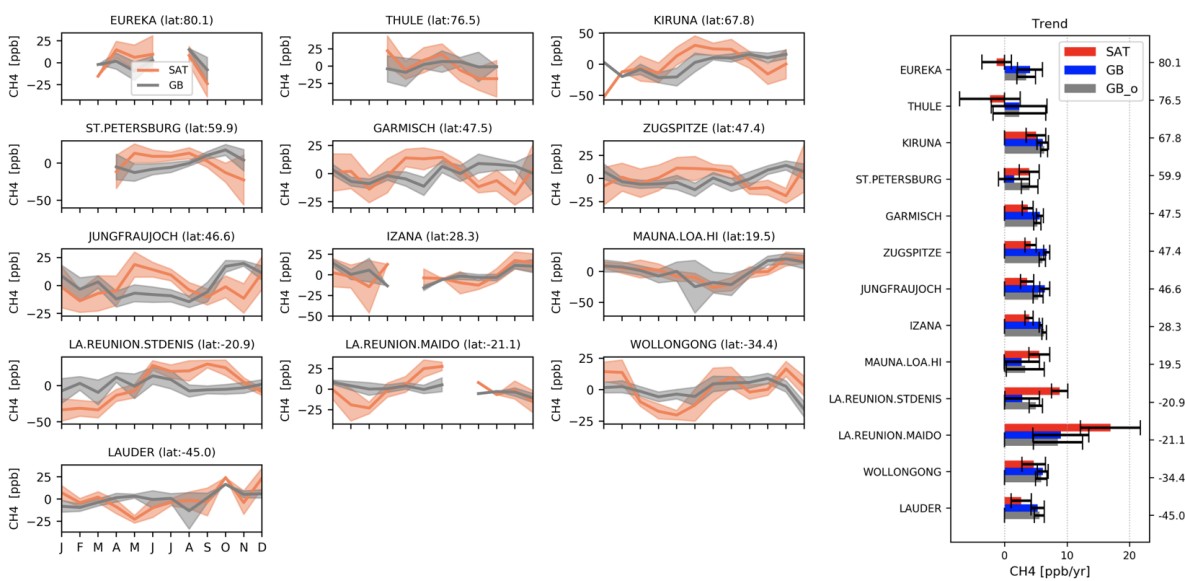

**Figure 21.** Same as Figure 18, but for LMD and NDACC measurements.

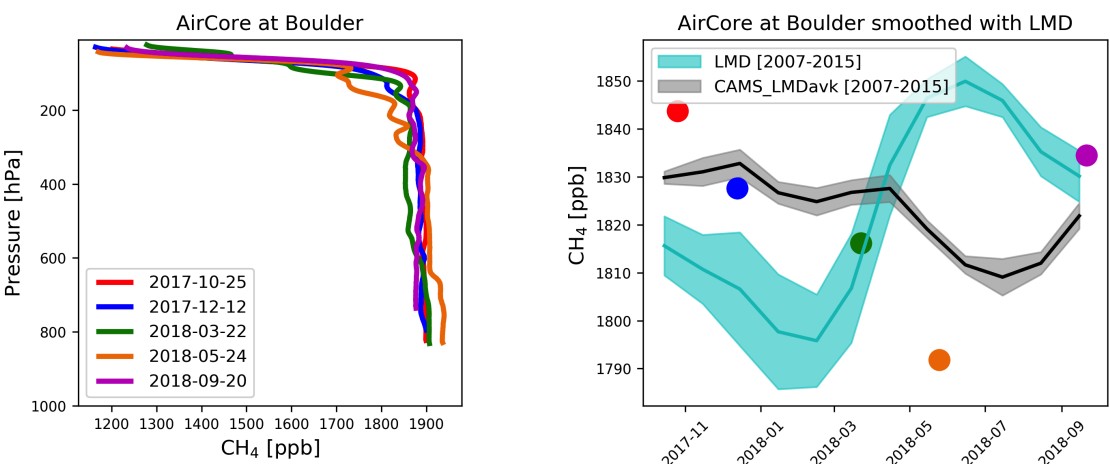

**Figure 22.** Left: the AirCore CH₄ profiles at Boulder between October 2017 and September 2018. Right: the time series of the mtCH₄ derived from the AirCore measurements with the smoothing correction using LMD weighting function, together with the mtCH₄ seasonal variation derived from LMD measurements between July 2007 and June 2015 with a constant shift using the mean of the 5 AirCore measurements. CAMS_LMDavk corresponds with CAMS model data at Boulder smoothed by the LMD weighting function.