# Peer review of "Independent validation of IASI/METOP-A LMD and RAL CH4 products using CAMS model, *in situ* profiles and ground-based FTIR measurements"

_Atmospheric Measurement Techniques, 2023_

## Author Comment (AC1)

**Reply to comments raised by Referee 1**

*The original comments are written in plain text, our replies are given in italic*

1) General comments

This study compares the accuracy of two algorithms in determining methane (CH4) concentrations from IASI sensor data. The retrieval results with the two algorithms are compared with each other and with independent observations and model data using various methods.

It is crucial to validate IASI CH4 data comprehensively for scientific use, and this paper's scientific significance is high. Therefore, I believe this paper's content deserves to be published in AMT.

However, the structure and description of this paper are somewhat confusing. In particular, the intercomparison part in Section 4 needs a more organized structure. The "short summary" subsection should be included in each intercomparison subsection, with tables, to aid readers in comprehending the validation results.

Also, regarding the gap in the data for 2013, it is important to include clear communication in the data section that this issue exists. Then, a comparative validation distinguishing between before and after data, or at least an intercomparison between before and after RAL and LMD, should be performed to show the presence or absence of an impact.

In addition, several figures need more legends or are unreadable. Please improve visibility.

Considering the above, I recommend that this paper should be published after examining the issues and making necessary revisions.

*I would like to thank the reviewer for the useful comments. A common comment of both reviewers pertains to the structure of the paper and upon reflection, I have to concur with their observations.*

*Particularly, the comparisons with independent reference data are impacted by the various temporal instabilities that are present in both datasets be it gradual or sudden. In the current structure, these temporal issues are discussed in depth after the in situ comparisons (in part because the temporal hiatus was discovered during the project). We therefore suggest a change in the structure of the paper in which we first focus on the LMD and RAL side-by-side comparisons using CAMS as an intermediate, including a figure that shows the LMD-RAL global bias distribution (as the bottom row of current figure 4) as a function of time (see figure below as an example). Then, after a discussion on the various temporal dependencies that are at play, we*

*will turn to the independent reference data, where we break up the RAL data into two segments due to its bias shift in 2023.*

[Figure]

*Figure 1: The difference in LMD and RAL_LMDavk biases relative to CAMS for different months (columns) and years (rows)*

2) Specific comments

・Page3 Line20 (P3L20): about IFOV of RAL measurements

Why use the IFOV with the highest brightness temperature among the four IFOVs instead of the average of the four?

*The IFOV with highest BT is assumed to be the one with least amount of potential cloud contamination. The RAL team therefore saw no obvious benefit in averaging over the four IFOVs.*

・P8L19: about vertical sensitivity of LMD and RAL_LMDavk

You note that there is a difference in altitude concerning the sensitivity of the two, but how much difference would this be in mtCH4? Can mtCH4 be evaluated using, for example, climatological values?

*As can be seen in figure 2, the altitude range to which both products show significant sensitivity differs substantially with RAL having peak sensitivity at ~600 to 400 hPa and LMD between 400 and 200 hPa. Furthermore LMD's near surface sensitivity is practically 0. Smoothing RAL with the LMD averaging kernel* does *align them better but there still is a vertical shift in the peak sensitivity region (with LMD's peak at lower pressures). Of importance to note here is that therefore LMD is more sensitive to methane residing in the so-called UTLS (Upper Troposphere-Lower stratosphere) and it is just in this region that a steep decline in the CH4 concentrations takes place as one moves from the troposphere into the stratosphere. The difference between RAL and LMD mtCH4 (due to smoothing errors alone!) is thus highly dependent on the true atmospheric state and the location of the tropopause height. A significant hint as to the magnitude of this vertical sensitivity bias are the two bottom rows of Figure 4, where a direct comparison between the 2 mtCH4 products (3$^{rd}$ row) show a bias around -12 to -15 ppb, while if we compare them individually against CAMS their bias ranges between -1.7 to 0.4 pbb (4$^{th}$ row). Thus we can state with some confidence that the differences in vertical sensitivity between LMD and RAL_LMDavk still amounts to a ~10 ppb bias, but locally they can be much higher (as can be observed when comparing specific regions between rows 3 and 4 of Figure 4)*

*As to whether mtCH4 can be evaluated using climatological values, we can state that these can indeed be used for a qualitative evaluation to some extent (for instance to verify trends and large-scale spatial distributions), but we felt that in the context of this work a chemical transport model is a much more suitable dataset.*

・P8L23: about grid averaging

Why did you use a 1-degree grid for intercomparison? This corresponds to a 100 km grid on a spatial scale, which might be too large, considering that the source of CH4 is more localized than CO2.

*It is true that CH$_4$ sources are more localized and that, if we were to focus on issues such as emission hot-spot localization etc., a 1-by-1 degree resolution might be too large. However given that IASI's near surface sensitivity is relatively low, a fair degree of mixing/dilution has already taken place once air masses reach the altitude range where the satellite's sensitivity is at its optimum. Furthermore, in this article we focus on global larger scale phenomena (regional biases, long-term trends and seasonality) which do not require a high resolution approach.*

・P9L4 and P9L10-L17: about selection criteria

The time frame for simultaneous observation with IASI is currently at ±6 hours. However, I believe this interval may be too long, considering the horizontal and vertical transport of CH4. CH4 distribution can vary significantly over such a period, so it may not be accurate to consider the observation as genuinely simultaneous.

Additionally, comparing ground-based FTIRs to the satellite dataset is unfair due to the differences in time and longitude ranges for each comparison. Can the conditions be as consistent as possible?

*As with the previous answer, IASI's vertical sensitivity range dampens the effect of localized sources and a more relaxed collocation approach can thus be justified. That said, finding the right collocation criteria is always a compromise between having a large enough data sample and minimizing the introduction of additional biases into the comparison. Using our fairly relaxed collocation criteria, we indeed cannot guarantee that on a station to station basis, no biases are introduced into the system. It is therefore important to look at the network dataset as a whole, certainly when focusing on larger scale phenomena.*

・P9L5: about the mean of the satellite values

How many satellite data points are usually averaged? And how large is the variability (stdv) of these data?

*The number of individual RAL satellite data points that are typically averaged range between 1 and 24, with a mean of 8.1. The stdv of the RAL co-located CH4 is about 12.4 ppb. For LMD the number of averaged data ranges between 1 and 15, with a mean of 3.9. The stdv of the LMD co-located CH4 is about 9.1 ppb.*

・P10L21: about eq.(7)

Please tell me how this equation was derived. Also, please explain what $C'_{r,R}$ on the left side represents.

*$C'_{r,R}$ is the retrieved RAL XCH4 where the impact of its own a priori has been replaced by the TCCON a priori. The equation stems from Rodgers [Rodgers, C.D. (2000) Inverse Methods for Atmospheric Sounding: Theory and Practice. World Scientific, River Edge.] where the retrieved quantity $C_r = C_{ap} + A(x - x_{ap})$, with x being the true state profile and $x_{ap}$ the a priori profile and $C_r$ and $C_{ap}$ the retrieved and a priori mole fractions respectively. The equation directly follows from when you were to calculate the difference between two $C_r$ using 2 different a priori profiles. We will clarify this in the text.*

・P11L9, L24, and L28: about equations (9), (11), (12)

What does the right-hand side of equations (9), (11), and (12) represent, respectively? Please add an explanation.

*Equation 9 is as equation 7 (replacing the influence of the a priori). Equations 11 and 12 pertain to the LMD product which does not include averaging kernel information. Therefore it contains no a priori information and no transformations regarding an a priori (as with equations 9 and 7) is required. Instead, it employs a sensitivity profile with a weighting function wf (see equation 1).*

*However in the equations there looks to be a space between w and f where there should be none as it is one parameter, which leads to confusion. This will be fixed and information will be added in the text.*

・P13L5-L6: about internal consistency

Why was the internal consistency only checked in October 2014? Could there be differences depending on the season, especially summer and winter? Also, is there any impact of the 2013 gap?

*The internal consistency has been checked only for October 2014 because the main objective here was to analyze the possible impact of slight instrumental inter-pixel calibration defects in the L2 data. There is no need to perform seasonal analysis to investigate this kind of systematic effects. The number of analyzed pixels (~90000) on a global scale is large enough to assess the systematic instrumental biases.*

*The 16th of May 2013 gap is a result of a change in the ground configuration for the spectral calibration introducing a band dependent PSF (Point Spread Function). This modification improved, among others, the interpixel spectral calibration especially in Band 2 as shown in the following figures. However, we do not expect these near negligeable interpixel spectral departures to have a significant impact on Level 2 retrieval since these departures especially in Band 2 are much weaker than the specification for IASI instruments ($\Delta v/v = 2.10^{-6}$ ).*

[Figure]

*Figure 2 : Residual inter-pixel relative spectral calibration ($\Delta v/v$) for the orbit of 2012/08/02 04h24 (Jacquette et al., 2013)*

[Figure]

*Figure 3 : Residual inter-pixel relative spectral calibration (Δv/v) for the orbit of 2013/09/12 04h21 (IASI quarterly performance report from 2013/09/01 to 2013/11/30)*

*E. Jacquette, B. Tournier , E. Péquignot , J. Donnadille , D. Jouglet  , V. Lonjou  , J. Chinaud , C. Baque , L. Buffet : IASI spectral calibration monitoring on MetOp-A and MetOp-B, 3rd IASI conference, 4-8 February 2013, Hyères, France*

・P13L30: IFOV selection

The authors claim that "RAL selects the best IFOV among 4 of them.", but in the previous section they mention "the one with the highest brightness temperature". Why is this the best?

*The IFOV with highest BT is assumed to be the one with least amount of potential cloud contamination. We will mention this in the revised paper.*

・P14L16: about SAT

What is "SAT"? Does SAT mean RAL and/or LMD measurements?

*SAT pertains to satellite and hence either RAL or LMD (smoothed and unsmoothed). This will be clarified in the text.*

・P15L13-L14: about the correlation between HIPPO and RAL measurements

What is the correlation between the two values obtained from the best fit?

*I'm not sure what you are implying with 'best fit'. Lines 13-14 only mention the linear fit in the correlation plot. If this is the fit you allude to, then by definition R=1.*

· P15L15-L18: comparison with IAGOS

Why is the correlation coefficient with IAGOS lower than other comparisons? Is such a low correlation coefficient due to the spatial bias of IAGOS measurements?

*There are many factors at play here. One is indeed that the IAGOS data is more geographically dispersed compared to HIPPO and Aircore and its descend and ascend profiles are typically located at or near urban centers, which could imply that local biases have a stronger effect on the IAGOS data. Another important aspect is that the vertical range covered by IAGOS is more restricted compared to HIPPO, and even much more so compared to Aircore. This entails that it relies more on the extrapolation of its data, yet another source of uncertainty. We will add this to the discussion of the results.*

· P17L24-P19L9: about subsection 4.7

This subsection repeats previous statements, making it redundant. It would be more useful to list each table in a related subsection.

*While we do think that a short summary is of use after each section, you are correct that the information presented under 4.7 goes beyond the scope of a summary and much of its content (and tables) should be presented under the related subsections. This will be applied to the restructured paper.*

· P19L10-P19L15: about discontinuity in RAL L2 data in mid-2013

Does the discontinuity in mid-2013 also affect the intercomparisons made in Section 4? For example, the comparison with AirCore and IAGOS was made using all data before and after the discontinuity. Would there be a difference in bias before and after?

Also, on page 23, the authors only compared partial columns between RAL and CAMS in 2012. Is there any difference in partial column bias before and after 2013?

*This is indeed a valid point. In the current structuring of the paper, the discussion of the temporal effects come after the comparisons with reference data. This will be changed and comparisons will then be evaluated prior and post May 2013.*

*The partial column comparisons between RAL and CAMS were carried out for all years (for instance, shown below is the upper-lower RAL bias for January 2014). While there might be*

*small differences in the absolute values, the observations and conclusions drawn from these comparisons remain the same. We will add a line to convey this message.*

[Figure]

*Figure 4: upper-lower RAL qCH4 bias for January 2014*

・Figure 5, 6, 7, 12 and 13:

The legend should be relocated or resized to avoid overlapping the plots. In Figure 13, please add a legend.

*This will be done*

・Figure 5, 6 and 7:

Could you please explain the meaning of the dotted lines in the correlation diagram?

*This is a simple linear fit (without forcing the fit to go through (0,0)). We will make this more clear in the figure text.*

3) Technical corrections

P2L9: andh => and

P4L23: (about 30 km; (Karion et al., 2010)) => (about 30 km) (Karion et al., 2010).

P5L3: CAMS model) as => CAMS model as

P7L9: in (Massart et al., 2014) => in Massart et al. (2014)

P9L20: According to (Rodgers and Connor, 2003) => According to Rodgers and Connor (2003),

P10L4: "$c_i$" in bold should be plain.

P10L7: respectively; => respectively,

P10L7: XCH4 => XCH4. (need period)

P15L23, L28 and L29: XCH4 => qCH4

Table 1: ParkFalls => Park Falls?

Figure 1: Rikubetsu should be "TCCON (yellow)", not "NDACC (green)".

Figure 4: "The last three rows show …" maybe "The last two rows show …".

Figure 6: In the x- and y-axis labels, "XCH4" should b

e "qCH4".

*Thank you for finding these errors. The suggested technical corrections will all be implemented*

---

## Author Comment (AC2)

**Reply to comments raised by Referee 2**

Comment on "Independent validation of IASI/METOP-A LMD and RAL $CH_4$ products using CAMS model, in situ profiles and ground-based FTIR measurements" by Bart Dils et al.

General:

This paper describes the validations of two methane products from IASI/METOP-A satellite sensor. This work is important, but I couldn't understand why the authors think that it is enough to use RAL data without correction. They pointed out there is a discontinuity in mid-2013 and compared some trends in Section 5.4. The corrected trend (5.6 ppb/yr) is significantly higher than the original one (4.2 ppb/yr) and the difference is larger than the standard deviation of original one. I think the trend analyses in Section 5.1 and 5.2 should be done using corrected data, or at least separately for the periods before and after mid-2013. Furthermore, the discontinuity should affect the results of validation in Section 4 but there is no explanation. The validation of RAL should also be done using corrected data, or separately for the periods before and after mid-2013.

The paper should be published after major revisions.

*I would like to thank the reviewer for the useful comments. A common comment of both reviewers pertains to the structure of the paper and upon reflection, I have to concur with their observations.*

*Particularly, the comparisons with independent reference data are impacted by the various temporal instabilities that are present in both datasets be it gradual or sudden. In the current structure, these temporal issues are discussed in depth after the in situ comparisons (in part because the temporal hiatus was discovered during the project). We therefore suggest a change in the structure of the paper in which we first focus on the LMD and RAL side-by-side comparisons using CAMS as an intermediate, including a figure that shows the LMD-RAL global bias distribution (as the bottom row of current figure 4) as a function of time (see figure below as an example). Then, after a discussion on the various temporal dependencies that are at play, we will turn to the independent reference data, where we break up the RAL data into two segments due to its bias shift in 2023.*

[Figure]

*Figure 1: The difference in LMD and RAL_LMDavk biases relative to CAMS for different months (columns) and years (rows)*

Comments and questions:

*All technical suggestions will be implemented in the revised document. More specific questions are answered below.*

Abstract

p1, l17

 The long-term stability --> The long-term trend

*This will be corrected*

Section 1

p2, l5 and 8

IPCC, 2013 --> It is better to refer newer report (IPCC, 2021)

*This will be corrected*

Section 2

p6, l8

largly --> largely

*This will be corrected*

p7 2.4

It is better to describe the original spatial resolution of CAMS because it was averaged onto a 1-degree latitude and 1-degree longitude grid before comparison.

*This will be implemented*

Section 3

p9, l8-9

Please discuss on the impact of this difference to the validation results in Section 4.7.

*This section will be impacted by the planned restructuring of the paper. However we will add a paragraph on discussing the potential impact of the sensitivity differences on their respective validation results.*

Section 4

p13, l4

What is 'pixel'? Is it mostly the same as IFOV? Please explain it in Section 2.

*The paragraph bottom of page 13 (line 29-31) states the difference between RAL and LMD regarding their use of the IFOV with the highest Brightness temperature vs. the average thereof. So what we further indicate as pixel depends on the algorithm. A line will be added so to better define this term, and if possible the ambiguous term itself is replaced by measurement.*

p14, l28

June --> July

*This will be corrected*

p15, l24

 … and IAGOS … : IAGOS isn't used for Figure 6. This is misleading.

Re-word in the revised version.

*This will be corrected*

l28-29

 XCH$_4$ generally …: What does this sentence mean? Are there any relations to the validation results?

*I agree that the term is too vague. The phrase points to Figure 6. Where you see a clear increase in the 0-6 km layer as you move from the equator to Northern latitudes. The 'general' alludes to the fact that this of course does not hold true from an individual measurement to another individual measurement basis. We will rephrase it as such:*

*The 0-6km partial column, figure 6 (top), shows a consistent qCH4 upward trend with latitude in the Northern Hemisphere. For the 6-12km partial column, figure 6 (bottom), two qCH4 concentration peaks can be observed around 35°N and 75°N.*

p16, l32

 Figure 8 --> Figure 8 (left)

*This will be corrected*

p17, l7

 Figure 8 --> Figure 8 (right)

*This will be corrected*

l10

 Tule --> Thule

*This will be corrected*

l11-13

Why this sentence is put here and the content is different from that described in l2-3.

*This is indeed an oversight on our behalf. An erroneous statement was left in the text next to its correction. The strongest biases are observed in spring (when ice starts to thaw) and therefore the last sentence is redundant and will be removed.*

p18, l3-6

… temporally …: Does HIPPO observation limited to some season? What figure in the top row of Figure 4 should be referred? This sentence is too vague.

*The HIPPO measurements were taken on a campaign basis and each campaign was by default limited in time. All campaigns combined do make sure that all seasons are covered. HIPPO I took place in January 2009; HIPPO II in October-November 2009; HIPPO III in March-April 2010, HIPPO IV in June- July 2011, and HIPPO V in August-September 2011. (These period descriptions will be added to the HIPPO description)*

*However, this sentence specifically alludes to the observation that when we look at the subset of HIPPO comparisons in figure 7 where the bias was the strongest we found that they matched the location and timeframe that corresponded with our earlier assessments. We will make this more exact, denoting the locations and timeframes in question.*

Section 5

p20, l11

It is found that … --> At land regions, it is found that ...

*This will be corrected*

l33-p21, l1

What about Maido?

*Maïdo only started NDACC measurements since 2013 (so only a short time coverage), therefore concerning the issue discussed, it does not provide a clear message and therefore it is not listed here.*

p21, l25-27

What is the definition of 'similar' and 'different'? Why Wollongong is 'similar' but Lauder isn't categorized?

*Here 'similar' means: the patterns of their seasonal variations are the same. 'different' means the patterns of their seasonal variations are different or opposite. We will reword the sentence to make clear it pertains to the seasonal cycle and/or phase. We listed stations as examples, not as an extensive list. In this case however we probably listed too many stations as examples giving the impression that it indeed should be a full list. We will make this clear in the revised document.*

Section 6

p23, l3

 Section 4.1 --> Section 4.3.1

*This will be corrected*

Section 7

p26, l8

 The long-term stability --> The long-term trend

*This will be corrected*

Many figures

*Thanks for bringing up these points regarding the figures. We will correct all mistakes and will work to make them clearer. As for figure size, we realize that we needed to make a compromise between avoiding overloading the paper with (larger) figures and disseminating enough information. We will work on improving their visibility.*

 The legends covered some of data plot. Move them not to cover the data.

 Size of the figure is too small.

Figure 1

There are many mistakes. For example,

 I couldn't find 'AirCore' mark at Sodankylä.

 I found 'NDACC' mark at Ny-Ålesund but Ny-Ålesund isn't listed in Table 2.

 I found 'NDACC' mark at Rikubetsu but Rikubetsu is listed in Table 1 (TCCON site).

Please check carefully.

*This will be corrected*

Figure 2 caption

 The solid data --> The solid line

*This will be corrected*

Figure 5, 6, and 7

 The correlation plots should be written with HIPPO values in horizontal axis. The values to be validated (RAL or LMD) should be in the vertical axis.

*This will be corrected*

Figure 5 caption

 the scatter plot between the RAL and HIPPO $XCH_4$.

--> the scatter plot between the RAL and HIPPO $XCH_4$ (right).

*This will be corrected*

Please add the explanation of the gray bar in the left figure, black bar, black solid line, and pink dashed line in the right figure.

*This will be corrected*

Figure 6 caption

Same as the comments for Figure 5.

*This will be corrected*

Figure 7 legend

 GB --> AIR

*This will be corrected*

Figure 14

 There is no error bar for Rikubetsu.

*There are only 8 co-located satellite-FTIR data pairs at the Rikubetsu site. Therefore the uncertainty is very large. In the revised version, we have removed Rikubetsu in Figure 14.*

---

## Author Response (AR1)

**Author's Response for amt-2023-237**

The main issue with the original version, as pointed out by the reviewers, was the handling of the RAL May 16th 2013 discontinuity problem. In our initial version we show the validation with reference measurements and afterwards, when discussing long term trends this issue was brought up and elaborated upon. In part this scheme followed the workflow within this project where this issue was identified after comparisons with said data. Unfortunately this of course leaves us with validation parameters that were performed on uncorrected data .

In the reworked version of this paper we reordered matters significantly, which now yields comparison parameters on corrected data.

Therefore, prior to any comparisons, a section 3.7 was added to the Methodology section in which we outline the May 16th 2013 issue with the RAL data, quantify the discontinuity and discuss several correction options.

Chapter 4 in the original paper discussed internal consistency, direct comparisons and comparisons with reference measurements. Chapter 5 then discussed the long-term trend. Chapter 6 then discussed the partial column differences in RAL and the seasonal cycle issues in LMD.

In this update we first discuss subjects that are internal to each of the products. This section contains a subsection on Internal consistency (previously and now 4.1) and one on the RAL partial column differences (now 4.2), which takes many paragraphs from the original version's 6.1 section (Discussion on two partial columns).

Section 5 is now called 'Direct Comparisons' in which we compare LMD to RAL, either directly or using CAMS as an intermediate. It discussed absolute differences (5.1, previously 4.2) and Long-term trends (5.2 previously 5.1). This section also contains a new figure showing the RAL vs LMD monthly averaged global biases for several years (instead of the example 2012 year only) and a more elaborate version of figure 21 (now Figure 12) that shows and discusses the differences in trend and seasonal cycle for LMD, RAL and RAL_LMDavk with respect to CAMS for several latitude bands (and now also for several May 16th discontinuity correction methods). This way, the reader is not only aware of the issue but has a better understanding of the potential impact when applying corrections prior to the actual reference measurement comparisons.

Section 6 then goes on to discuss the actual comparisons with reference measurements (in situ and FTIR). In this section, as with the previous section on long-term trends, this time RAL discontinuity corrections have been applied to the dataset. It contains information from the original sections 4.3 to 4.7 although some sub-chapters have changed order. Information in 4.7 (short summary, but (rightly) deemed too elaborate and repetitive by the reviewer), has to a large degree been transferred to the respective subsections making place for a more compact summary.

Section 7 (Discussion) corresponds with previous version Section 6 (although several paragraphs on the partial column analysis transferred to new section 4 and Section 8 Conclusions also corresponds with its namesake in the previous version.

Due to the different structure, there is also a considerable rearrangement of figures. These changes have been listed in the Table below, showing the new numbering, the old numbering and mayor changes (other than improvements of the layout).

| New nr | Original nr | Comment |
| --- | --- | --- |
| 1 | 1 | |
| 2 | 2 | |
| 3 | 18 | Also showing partial columns |
| 4 | 3 | |
| 5 | 19 | |
| 6 | 4 | |
| 7 | New | As previous figure last row but for multiple years |
| 8 | 10 | |
| 9 | 11 | |
| 10 | 12 | Now with discontinuity correction |
| 11 | 13 | Now with discontinuity correction |
| 12 | 21 | Now with impact of correction methods |
| 13 | 5 | Now with discontinuity correction |
| 14 | 7 | |
| 15 | 6 | Now with discontinuity correction |
| 16 | 8 | |
| 17 | 9 | |
| 18 | 14 | Now with discontinuity correction |
| 19 | 15 | Now with discontinuity correction |
| 20 | 16 | |
| 21 | 17 | |
| 22 | 20 | |

Also all suggested technical corrections and clarifications, to the best of our abilities, have been added to the article.

Finally, I want, again, to thank the reviewers for their helpful suggestions and I hope that these changes meet their expectations.

---

## Referee Report (RR1)

Revised version

1) General comments
This study validates the accuracy of methane (CH4) concentration data from IASI sensor data retrieved with two retrieval algorithms (RAL and LMD). The data from RAL and LMD are compared with each other, with independent observations, and with model data, and their accuracy is examined not only by direct comparison of values but also by focusing on seasonal variations and trends.
The revised version presents the validation results and carefully summarizes the accuracy and problems with each algorithm's data. These are significant findings that are indispensable for the use of IASI CH4 data. Therefore, it is well worth publishing in AMT. In places, there are errors of citation in not a few places in the revision, but they are all minor issues.
In conclusion, I recommend that this paper should be published after making very minor corrections.

2) Specific comments
P14L24-29: edge viewing value vs. nadir value
Why is the value for the edge viewing in case mtCH4 LMD larger than that for the nadir viewing, but the nadir value is larger in case XCH4 RAL?

P19L24-25: "Both results are close to the mtCH4 annual growth of 5.3 ppb/yr ···."
What does "Both results" refer to? RAL_LMDavk and RAL_LMDavk (2 periods)?

P21L10-17: in Section 6
It would be better to clearly state in this part that the RAL data used here is bias-corrected.

3) Technical corrections
・Please unify abbreviations in the text.
 NLIS (P1L4, P2L22) or NILS (P3L19, P4L2, P4L9, P7L8)
 DOFs (P3L29, P6L7, P21L20) or DOFS (P15L15, P29L13) or DOF (P15L23 and L24, P16L13) or DOF's (P30L16)

P4L23: "((Karion et al., 2010))" should be "(Karion et al., 2010)".

P5L12: IAGOS CARIBIC should be "IAGOS-CARIBIC".

P11L10: "((Rodgers and Connor, 2003))" should be "(Rodgers and Connor, 2003)".

P13L11: "A0" and "A1" should be "$A_0$" and "$A_1$", respectively.

P16L16 and P18L8: "mayor" may be "major".

P18L33-34: References on TransCom and Carbon Tracker are needed.

P19L22: "Table 6" should be "Table 3".

P22L30: "Table 3" should be "Table 4".

P24L5: "Table 4" should be "Table 5".

P24L12: "/textitin" may be "in".

P28L26: "Table 5" should be "Table 6".

P30L19: "Figure 11" should be "Figure 10".

Table 6: The values for LMD differ from those in the text. Please check once.

Table 7: This table is not quoted in the text. Please delete the table or quote it appropriately in the text.

Figure 6: In the figure caption, "-20 deg to 20 deg" should be "-60 deg to 60 deg".

Figure 11: In the caption, "Figure 12" should be "Figure 10".

Figure 14: In the figure caption, "Figure 10" should be "Figure 13".

Figure 15: The two plots (6-12 km and 0-6 km) on the left should be replaced.

Figure 15: Since N is not used in the figure, the figure caption does not need to explain it.

---

## Author Response (AR2)

**Reply to referee #1**

First of all I would like to thank the reviewer for the thorough evaluation of our paper. All suggested technical corrections have been implemented. Furthermore there were some additional questions which I have reiterated below with our responses in italic

P14L24-29: edge viewing value vs. nadir value Why is the value for the edge viewing in case mtCH4 LMD larger than that for the nadir viewing, but the nadir value is larger in case XCH4 RAL?

*Given the very different natures of both retrieval methods, it is very difficult to truly assess the impact of the viewing angles upon the final products and why they differ so much between algorithms. This is a topic of further investigation by the algorithm development teams.*

P19L24-25: "Both results are close to the mtCH4 annual growth of 5.3 ppb/yr …."

What does "Both results" refer to? RAL_LMDavk and RAL_LMDavk (2 periods)?

*This is an error on our behalf and it should be a singular expression: This result…*

P21L10-17: in Section 6

It would be better to clearly state in this part that the RAL data used here is bias-corrected.

*This has been implemented in the beginning of Section 6*

*All the technical corrections listed below have been implemented*

3) Technical corrections

・Please unify abbreviations in the text.

NLIS (P1L4, P2L22) or NILS (P3L19, P4L2, P4L9, P7L8)

DOFs (P3L29, P6L7, P21L20) or DOFS (P15L15, P29L13) or DOF (P15L23 and L24,

P16L13) or DOF's (P30L16)

P4L23: "((Karion et al., 2010))" should be "(Karion et al., 2010)".

P5L12: IAGOS CARIBIC should be "IAGOS-CARIBIC".

P11L10: "((Rodgers and Connor, 2003))" should be "(Rodgers and Connor, 2003)".

P13L11: "A0" and "A1" should be "A0" and "A1", respectively.

P16L16 and P18L8: "mayor" may be "major".

P18L33-34: References on TransCom and Carbon Tracker are needed.

P19L22: "Table 6" should be "Table 3".

P22L30: "Table 3" should be "Table 4".

P24L5: "Table 4" should be "Table 5".

P24L12: "/textitin" may be "in".

P28L26: "Table 5" should be "Table 6".

P30L19: "Figure 11" should be "Figure 10".

Table 6: The values for LMD differ from those in the text. Please check once.

Table 7: This table is not quoted in the text. Please delete the table or quote it appropriately in the text.

Figure 6: In the figure caption, "-20 deg to 20 deg"should be "-60 deg to 60 deg".

Figure 11: In the caption, "Figure 12"should be "Figure 10".

Figure 14: In the figure caption, "Figure 10"should be "Figure 13".

Figure 15: The two plots (6-12 km and 0-6 km) on the left should be replaced.

Figure 15: Since N is not used in the figure, the figure caption does not need to explain it.

**Reply to Referee #2**

Again, I would like to thank the reviewer for his in depth and thorough report. All the suggested technical corrections have been implemented, and where questions were put forth, we replied to these in the text below (in italic).

On a more general note, there was some confusion as to the nature of the so-called core stations. We have tried to clarify this in the following paragraph as follows:

*Here we compared the LMD and RAL methane data products with ground-based remote sensing data from the TCCON and NDACC networks. As with the in situ comparisons, a +6.7 ppb correction has been applied to the RAL post May 16th 2013 data. Also note that there is substantial difference in the time-periods covered by the individual stations. For TCCON, the stations that cover almost the entire time period (less than 2.5 years of missing data), and hence-forward referred to as core stations, are: Sodankyla, Bialystock, Orleans, Garmish, Park Falls, Lamont, Izana, Darwin, Wollongong and Lauder. Other stations on the other hand have noticeably shorter coverages (Rikubetsu and Edwards for instance have less than 2 years of co-located measurements). For NDACC, all stations are listed as core stations as they cover quasi the entire timeperiod, apart from Maïdo (2.5 years of data), which is excluded. Note that both Mauna Loa and St-Denis feature some large (> 1 year) data gaps in their time series and that even with a long time span, the number of co-located data pairs may differ greatly between stations. For instance at high latitude sites (Eureka, Thule) annual gaps occur in the dataset during wintertime (see Figures 16 and 17). For RAL, the amount of pre- versus post-discontinuity data largely determines the magnitude of the impact of the applied discontinuity-correction and therefore when comparing average*

*overall long-term trends we have restricted ourselves to the so-called core stations which cover a substantially long time period as listed above.*

Comments and questions:
Abstract
p1, l14
difference is --> differences are
Section 2
p4, l3
AMSU --> Advanced Microwave Sounding Unit (AMSU)
p4, l23
((Karion et al., 2010)) --> (Karion et al., 2010)
p5, l2
'returned either over the Eastern Pacific, or over the Western Atlantic'
Is it true? I couldn't find Western Atlantic path in Figure 1.

*This should of course be Western Pacific*

Section 3
p8, l8
weighing --> weighting
p13, l11
A0, A1 --> A_0, A_1 (subscript)
p14, l2
60°N-60° correction --> 60°N-60°S correction
Section 4
p16, l15
short summary --> Short summary
p16, l16
mayor --> major
p16, l17
interpixel --> inter-IFOV
Section 5
p17, l25-27 and l30-32
The following two sentences maybe the same:
'In October, this positive bias band has shifted towards the Southern hemisphere. Strong negative biases are observed over the Canadian Boreal forests.'
'In October a positive latitudinal bias belt between 10°S and 30°S can be observed over land and sea, while outspoken negative biases are visible at high Northern latitudes'

*They are indeed and have been merged*

p18, l8
mayor --> major
p19, l22
Table 6 --> Table 3

p19, l24
'Both results ...'
I think you only compare one result (RAL_LMDavk (2 periods)) with that by LMD.

*Yes, this phrase is now in a singular form*

p19, l30
seasonal cycles --> seasonal cycle amplitudes
p20, Table 3 caption
RAL_LMDavk (corrected) --> RAL_LMDavk (2 periods)
p21, l1
short summary --> Short summary
Section 6
p21, l16
United States and Finland --> United States, New Zealand, and Finland
p22, l30
Table 3 --> Table 4
p23, l9
in the vertical range of 6-12 km --> Please remove.
p23, l16
You changed the order from
'IAGOS and RAL measurements'
to
'RAL measurements and IAGOS'.
If you want to use the order of
"to be validated" and "reference measurements",
it is better to use this order throughout the paper.

*This has been implemented throughout the paper*

p23, l18
+6.6 ppb --> +9.6 ppb
p24, l5
Table 4 --> Table 5
p24, l10
comparisons --> Comparisons
p24, l12
/textitin situ --> in situ
p24, l20
post-discrepancy --> post-discontinuity
p24, l20
'an obvious impact on the magnitude of the impact of the applied discontinuity-correction'
This is confusing because you use two 'impact'. Please consider revising.

*Reworded: For RAL, the amount of pre- versus post-discontinuity data largely determines the magnitude of the impact of the applied discontinuity-correction and therefore when comparing average overall*

*long-term trends we have restricted ourselves to the so-called core stations which cover a substantially long time period as listed above*

p25, l26
by --> maybe 'in comparison with' is better
p26, 5-8
'Again looking … other stations.'
I couldn't understand what you want to say from the latter part of this sentence (after 'apart from' phrase).

*Reworded: Again looking at the long-running core stations, the RAL-TCCON long-term trend bias difference, ranging between -2.72±0.62 ppb/year (Lamont) and -0.47±0.46 ppb/year (Lauder), shows little latitudinal dependence apart from the observation that Southern hemisphere RAL-TCCON trend differences are slightly smaller, compared to those observed at Northern hemisphere stations.*

p27, l3
What is 'core station'?
p27, l6-7
'but there is unfortunately very little data within this range and the variability outside this range is considerable.'
I think 'this range' means 40°S-40°N. There are 10 stations within this range and 8 stations outside this range from Figure 20. I couldn't understand why you said 'very little data' without the definition of 'core station'.

*The definition of core station has been added and section has been reworded: Overall the bias differences are more outspoken compared to the RAL-TCCON trend differences. We also typically find the strongest negative LMD-TCCON trend biases outside the 40°S-40°N range (-4.48 ppb/year at Lauder, 3.76 ppb/year at Sodankyla), but the variability within and outside the 40°S-40°N range is considerable. For instance, the trend difference at Orleans (48°N, -1.09 ppb/yr) is smaller than that observed at Lamont (36.6°N, -2.36 ppb/yr).*

p28, Table 7
The numbers misplaced. For example, 4.84 ppb is the RAL mean at TCCON sites.
Also, there is no description of the Table 7 in the text.

*This has been added:*

*Table 7 below shows the averaged (over all core stations) long term trends of both LMD and (bias-corrected) RAL and their corresponding co-located NDACC and TCCON mtCH4 timeseries. Overall, on average, both LMD and RAL underestimate 5 the long term trend. Also immediately apparent is the far greater standard deviation on the trend for LMD, compared to RAL, indicating stronger station-to-station variability.*

p28, l26
Table 5 --> Table 6
p29, l8
initial a priori profile --> a priori profile
Section 7

p30, l9-10
'not insignificant (but not impossible either)'
Is this means that significant and possible? There are many unclear expressions throughout the paper.

*Reworded: Changing the CAMS UTLS transition region resulted in significant changes in the observed biases with HIPPO. However in most cases, the biases increased instead of decreased and in the rare cases the comparison improved, upward shifts of up to 4 km of the transition region were required.*

Section 8
p31, l14
discrepancy --> discontinuity
p31, l17
Where the value '1.3 ppb' come from? ('1.1 ppb' maybe come from 15.2 – 14.1 in the Table 4 but I couldn't find 1.3 ppb.)

*This was an error, the phrase has been reworded: For AirCore these differences in stdv are small (15.2 ppb for LMD vs. 14.1 ppb for RAL).*

p31, l24
the long-term stability --> the long-term trend
p31, l27-28
'The L1 … now using.'
Is this for the newer version than RAL v2.0 that wasn't analyzed in this paper? If so, please clarify it. Instead of this information, you should describe the result with discontinuity correction.

*Only a very preliminary analysis of an intermediate update was performed. We removed this sentence to avoid confusion.*

p32, l3-4
'Improvements … reduced.'
Is this also for the newer version? Are you already analyzed it?

*See above*

Figure 1
There are 'AirCore' mark in the Atlantic Ocean. What is this?

*Unfortunately, in the AirCore datafile one entry was tagged with the wrong geolocation. This has been fixed. No impact on our further AirCore analysis was observed.*

Figure 3 caption
'for the -20° to 20° …'
The latitudes should be -60° to 60° and there are 4 graphs, not two.
Figure 6 caption
The last three rows --> The last two rows
Figure 8 caption
RAL (b) RAL_LMDavk (d) --> RAL (b), and RAL_LMDavk (d)
between LMD and RAL_LMDavk (e), between RAL and RAL_LMDavk (f)

--> between LMD and RAL_LMDavk (e), and between RAL and RAL_LMDavk (f)
Figure 12 caption
2013 timeserie. --> 2013 timeseries.
Figure 13 right figure
Is the dashed line correct? You changed x-axis and y-axis but the dashed line doesn't so change.
Figure 13 legend
( +6.7 ppb discontinuity corrected) --> remove or (no data after May 2013)
N is the number … --> now N isn't written in the figure. Please remove.
The colored dotted --> The dotted (now the line is black.)
'with forcing the fit to go through (0,0)'
Is it true? It looks only y=x line.

*It is indeed a y=x line, this has been corrected*

Figure 14 axis titles
CH4 --> mtCH4 is better
Figure 14 right figure
The dashed line also should be checked.
Figure 15
The positions of the left figures are wrong.
The dashed lines in the right figures also should be checked.
Figure 15 caption
Same as the comments for the Figure 13 caption
Figure 22 caption
There is no description for the CAMS_LMDavk.

*CAMS_LMDavk corresponds with CAMS model data at Boulder smoothed by the LMD weighting function.*